# Structural basis for the unique molecular properties of broad-range phospholipase C from *Listeria monocytogenes*

Nejc Petrišič [1,2], Maksimiljan Adamek [1], Andreja Kežar [1], Samo B. Hočevar [3], Ema Žagar [4], Gregor Anderluh [1] & Marjetka Podobnik [1] ✉

Listeriosis is one of the most serious foodborne diseases caused by the intracellular bacterium *Listeria monocytogenes*. Its two major virulence factors, broad-range phospholipase C (*Lm*PC-PLC) and the pore-forming toxin listeriolysin O (LLO), enable the bacterium to spread in the host by destroying cell membranes. Here, we determine the crystal structure of *Lm*PC-PLC and complement it with the functional analysis of this enzyme. This reveals that *Lm*PC-PLC has evolved several structural features to regulate its activity, including the invariant position of the N-terminal tryptophan (W1), the structurally plastic active site, $Zn^{2+}$-dependent activity, and the tendency to form oligomers with impaired enzymatic activity. We demonstrate that the enzymatic activity of *Lm*PC-PLC can be specifically inhibited by its propeptide added in trans. Furthermore, we show that the phospholipase activity of *Lm*PC-PLC facilitates the pore-forming activity of LLO and affects the morphology of LLO oligomerization on lipid membranes, revealing the multifaceted synergy of the two virulence factors.

The intracellular Gram-positive bacterium *Listeria monocytogenes* (*Lm*) is a causative agent of the foodborne infectious disease listeriosis[1]. Although it is a relatively rare disease, with over 23,000 cases reported worldwide in 2010, the mortality rate from listeriosis is high, reaching 20–30%[2]. Listeriosis therefore poses a significant health risk, particularly to vulnerable populations such as pregnant women, newborns, and immunocompromised individuals. It can lead to severe clinical manifestations such as gastroenteritis, meningoencephalitis, septicemia or miscarriage[1]. As it also infects wild and domestic animals, *Lm* poses a serious challenge to the food industry and agriculture worldwide[3].

After entering the host cell, the intracellular life cycle of *Lm* begins with escape from early phagosomal (primary) vacuoles into the cytosol, followed by replication in the cytosol and acquisition of actin-based motility that enables cell-to-cell spread of *Lm* packed in double-membrane (secondary) vacuoles, and finally escape from these vacuoles, restarting the cycle in neighboring host cells[4]. Infection by *Lm* is facilitated by a plethora of specialized bacterial effectors[5]. A pore-forming protein, listeriolysin O (LLO), is the major virulence factor, which facilitates phagosomal membrane disintegration[6,7]. LLO binds to lipid membranes where it oligomerizes and causes large-scale membrane disruption by forming structurally dynamic pores formed by the fusion of arcs and slit-like oligomers[8,9]. Two phospholipases C (PLCs), phosphatidyl inositol specific (*Lm*PI-PLC or PlcA) and broad-range (*Lm*PC-PLC or PlcB), facilitate vacuolar escape of *Lm* by cooperating with LLO in vacuolar membrane disruption[10,11]. In the secondary vacuole, listerial PLCs are essential for inner membrane dissolution and LLO for outer membrane disruption[12–14]. Interestingly, in some human epithelial cell lines, *Lm*PC-PLC enables bacteria to escape from vacuoles even in the absence of LLO[10,15]. Listerial PLCs may also be

[1]Department of Molecular Biology and Nanobiotechnology, National Institute of Chemistry, Ljubljana, Slovenia. [2]PhD Program 'Biosciences', Biotechnical Faculty, University of Ljubljana, Ljubljana, Slovenia. [3]Department of Analytical Chemistry, National Institute of Chemistry, Ljubljana, Slovenia. [4]Department of Polymer Chemistry and Technology, National Institute of Chemistry, Ljubljana, Slovenia. ✉e-mail: marjetka.podobnik@ki.si

involved in the recruitment of host factors such as protein kinases C beta-type and phospholipases C and D via their reaction product diacylglycerol (DAG), which contributes to membrane disintegration[16]. The presence of DAG itself can lead to instability of the membrane structure and its subsequent destruction[17].

Homologs of LLO and $Lm$PI-PLC are present in many Gram-positive bacteria[18,19]. Structural analyses of both proteins revealed unique adaptations to the intracellular environment at the molecular level[20,21]. Homologs of $Lm$PC-PLC, which belong to the family of $Zn^{2+}$-dependent phospholipases, have so far only been found in the genera *Bacillus* and *Clostridium*[22]. $Lm$PC-PLC has 40% amino acid sequence identity with the homolog from *Bacillus cereus* (*Bc*), *Bc*PC-PLC, and 22% with the catalytic N-terminal domain of PC-PLC from *Clostridium perfringens* (*Cp*), *Cp*PC-PLC[23]. Their active sites were proposed to harbor three $Zn^{2+}$ ions, coordinated by highly conserved $Zn^{2+}$-binding amino acid residues[24–27] (Supplementary Fig. 1). The crystal structures of *Bc*-[24,28] and *Cp*PC-PLC[25,29] show a globular α-helical folding of the phospholipase domain, while the structure of $Lm$PC-PLC has not yet been determined.

$Lm$PC-PLC is expressed as an inactive proenzyme with a 26 amino acid long propeptide[14]. After acidification of the phagosomal vacuole, the listerial metalloprotease Mpl[30] cleaves the propeptide from $Lm$PC-PLC, allowing translocation of the mature $Lm$PC-PLC through the bacterial cell wall into vacuole[30]. $Lm$PC-PLC exhibits broad substrate specificity, with the highest activities towards glycerophospholipids with phosphocholine, phosphoserine, and phosphoethanolamine head groups, but less towards phosphoglycerol or phosphoinositol head groups, and also hydrolyses sphingomyelin (SM)[26,31,32]. *Bc*PC-PLC and *Cp*PC-PLC also have broad activity towards phospholipids, but *Bc*PC-PLC lacks sphingomyelinase (SMase) activity[33,34]. Unlike $Lm$- and *Bc*PC-PLC, *Cp*PC-PLC is a two-domain protein[25]. PLC activity is located in its N-terminal PLC domain, while the C-terminal C2 domain is responsible for binding the enzyme to lipid membranes[35]. This confers hemolytic and cytotoxic properties to *Cp*PC-PLC[36], while *Bc*PC-PLC[37] and $Lm$PC-PLC[32] have no and very weak hemolytic activity, respectively.

LLO is a cholesterol-dependent cytolysin (CDCs)[38]. It acts preferentially on membranes with a high cholesterol (CHOL) content, 35-45 mol%[39]. Since $Lm$PC-PLC is encoded by the same virulence gene cluster as LLO and is co-expressed with LLO[40], it could facilitate the activity of LLO towards lipid membranes. This could be achieved by releasing the free CHOL receptor from complexes with other lipids by hydrolysis of lipid head groups, as shown for *Cp*PC-PLC[41]. However, data on the interplay between LLO and $Lm$PC-PLC at the molecular level are sparse[26,42].

Insights into the atomic structures and molecular mechanisms of action of virulence factors are a prerequisite for understanding the course of diseases and developing alternative solutions to curb them. Here we have determined the crystal structure of $Lm$PC-PLC, which reveals distinctive structural details in an otherwise conserved PC-PLC fold. This translates into unique properties of the listerial phospholipase, including significantly lower enzymatic activity compared to its homologs, which appears to be regulated by Zn ions, the structurally plastic active site of $Lm$PC-PLC and the tendency to form enzymatically impaired oligomers. Furthermore, we demonstrate that the phospholipase activity of $Lm$PC-PLC can be specifically inhibited by its propeptide added in trans. Finally, we show at the molecular level that the phospholipase activity of $Lm$PC-PLC facilitates lipid membrane disintegration by LLO and also influences the morphology of LLO oligomerization at the lipid membrane.

## Results

### $Lm$PC-PLC tends to oligomerize, but only the monomer is enzymatically active

$Lm$PC-PLC was produced as a recombinant protein in *Escherichia coli*. The protein eluted according to the size exclusion mechanism from the size exclusion column (SEC) only when using an optimized mobile phase composition (20 mM Tris-HCl pH 8.5, 500 mM NaCl). SEC chromatogram of $Lm$PC-PLC showed the presence of monomers, dimers and higher order oligomers (Fig. 1a, Supplementary Fig. 2a, b). The CD spectra of all three fractions were similar and showed a high proportion of the α-helical fold (Supplementary Fig. 2c). N-terminal sequencing of the 28 kDa bands from SDS-PAGE (Fig. 1a) confirmed the presence of mature $Lm$PC-PLC in SEC peaks 1 to 3 and in the 56 kDa band of peak 3, indicating the formation of an $Lm$PC-PLC dimer (Fig. 1b). The SEC-MALS analysis confirmed that $Lm$PC-PLC eluted from the SEC column in various forms, as monomers, dimers and higher order aggregates (Supplementary Fig. 2d). However, the enzymatic activity of the oligomeric fractions (SEC peaks 1 and 2) was significantly lower than that of the monomeric $Lm$PC-PLC in peak 3 (Fig. 1c), when a synthetic substrate for phospholipases, 4-nitrophenylphosphorylcholine (4-NPPC)[43], was used. The pooled fractions from peak 3 were reanalyzed on SEC and again split into multiple peaks containing $Lm$PC-PLC (Supplementary Fig. 3a), indicating that equilibrium was established between the monomer and the higher order oligomers. In the SEC analysis, we also found that the monomeric fraction of $Lm$PC-PLC (peak 3) eluted later than expected based on the calibration curve generated from the protein standards (Fig. 1a and Supplementary Fig. 3b). This indicates either a different hydrodynamic volume of $Lm$PC-PLC compared with the protein standard of similar molecular mass or interactions between $Lm$PC-PLC and the column matrix despite high salt concentration in the mobile phase. Interestingly, unlike $Lm$PC-PLC, where oligomer formation was observed by SEC, SDS-PAGE, and blue native-PAGE gels (Fig. 1a, b, Supplementary Fig. 2), the *Bc* and *Cp* homologs appeared to form only homogeneous monomeric molecular populations (Fig. 1a, d).

Concentrating of the pooled fraction of the SEC peak 3 of $Lm$PC-PLC to higher concentrations resulted in protein precipitation, which in turn prevented successful crystallization. $Lm$PC-PLC has two unique cysteines, C143 and C168, which are not present in *Bc* or *Cp* PC-PLC (Supplementary Fig. 1). The cysteines had a very limited effect on the oligomerization of $Lm$PC-PLC, as the addition of the reducing agent only partially dissolved the oligomers on SDS-PAGE and no appreciable difference was observed on the blue native-PAGE (Fig. 1d, e). Nevertheless, we replaced the cysteines with two serines or serine and lysine, the latter combination occurring naturally in *Bc*PC-PLC (Supplementary Fig. 1), to see if this would contribute to higher solubility. We made constructs with a single cysteine substitution, i.e. $Lm$PC-PLC[C143S] and $Lm$PC-PLC[C168S], or a double cysteine substitution, i.e. $Lm$PC-PLC[C143S+C168S] and $Lm$PC-PLC[C143S+C168K]. All mutant proteins behaved similarly to wild-type $Lm$PC-PLC upon purification, with a similar extent of oligomer formation (Supplementary Fig. 3c). $Lm$PC-PLC[C168S], $Lm$PC-PLC[C143S+C168S] and $Lm$PC-PLC[C143S+C168K] showed slightly reduced formation of oligomers (Fig. 1e, Supplementary Fig. 4a). The mutant proteins had comparable CD spectra to the wild-type enzyme (Supplementary Fig. 4b) and the SEC fraction corresponding to peak 3 of the wild-type enzyme retained enzymatic activity (Fig. 1f, g), tested with either the small synthetic substrate 4-NPPC or the lipid substrate 1-palmitoyl-2-oleoyl-glycero-3-phosphocholine (POPC) in multilamellar vesicles (MLVs). Of all the constructs, $Lm$PC-PLC[C143S+C168S] had the highest solubility and successfully produced crystals.

### The active site of recombinant $Lm$PC-PLC contains Zn and Fe ions

The activity of $Lm$PC-PLC was largely and specifically dependent on the presence of Zn ions (Fig. 2a). Therefore, to determine the crystal structure of $Lm$PC-PLC[C143S+C168S] (referred to as the crystal structure of $Lm$PC-PLC in the rest of the text), we used X-rays with a wavelength of 1.26 Å (Supplementary Table 1) near the absorption edge of Zn, based on previous studies suggesting that three Zn ions are in the active site of $Lm$PC-PLC[26] and the *Bc* and *Cp* homologs[25,27]. The crystal structure of $Lm$PC-PLC was solved using single-wavelength anomalous diffraction technique.

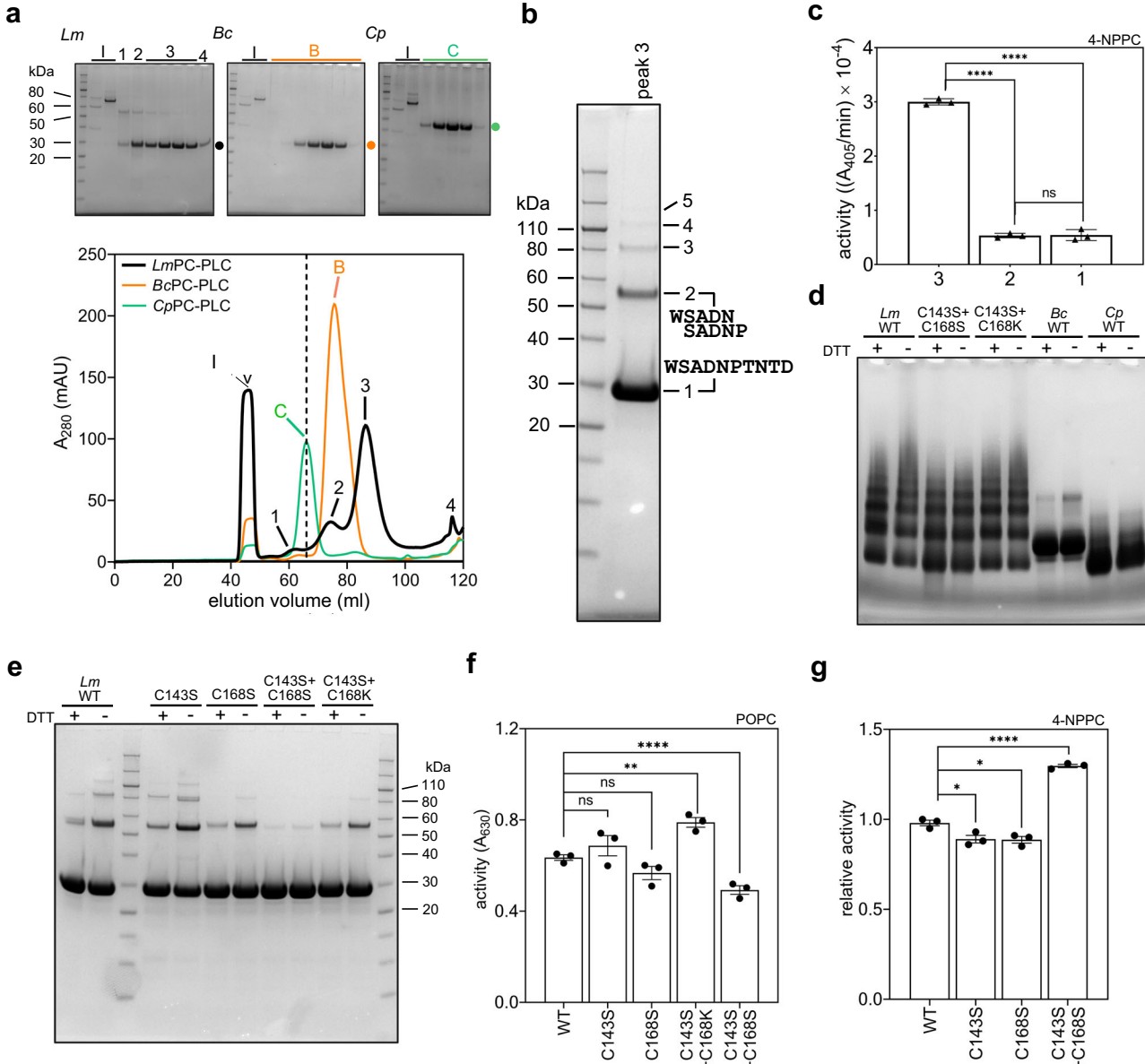

**Fig. 1 | *Lm*PC-PLC is prone to oligomerization. a** SEC chromatogram of the wild-type (WT) *Lm*-, *Bc*- and *Cp*-PC-PLC in 20 mM Tris-HCl pH 8.5, 500 NaCl. Dashed vertical line marks the expected elution volume for a globular 28 kDa protein (66 ml, Supplementary Fig. 3b). Above the SEC chromatogram: SDS-PAGE analysis of SEC fractions. 'V': fractions eluted in the void volume of the column. 'B': *Bc*PC-PLC, 'C': *Cp*PC-PLC. Black, orange and green dot on the right to each SDS-PAGE gel marks the position of bands corresponding to L*m*-, *Bc*- and *Cp*-PC-PLC. **b** SDS-PAGE of the concentrated SEC peak 3 from (**a**) of *Lm*PC-PLC, with results from the N-terminal sequencing. Numbers 1 to 5 indicate bands representing multiples of *Lm*PC-PLC monomer. **c** Enzymatic activity of the wild type (WT) *Lm*PC-PLC SEC peaks 1, 2, and 3 (500 µM, **a**) towards 1 mM 4-NPPC. **d** Blue native-PAGE of the WT *Lm*PC-PLC, cysteine mutants of *Lm*PC-PLC, WT *Bc*PC-PLC, and WT *Cp*PC-PLC, in the absence or presence of DTT (20 mM). **e** SDS-PAGE of *Lm*PC-PLC, WT and cysteine mutants, in the absence or presence of DTT (20 mM). **f** Enzymatic activity of WT *Lm*PC-PLC and cysteine mutants (50 nM) towards 100% POPC MLVs (4.5 mM). **g** Activity of WT *Lm*PC-PLC and cysteine mutants (500 nM) towards 4-NPPC (1 mM). Experiments in c, f and g were performed in 20 mM MES pH 6.5, 150 mM NaCl, 50 µM (**f**) or 500 µM (**c, g**) ZnSO$_4$ at 37 °C. Dunnett's multiple comparisons test was performed (**c, f, g**), ns: $P > 0.05$, *$P < 0.05$, **$P < 0.01$, ***$P < 0.001$, ****$P < 0.0001$. Source data and gel replicates (**d, e**) are provided as a Source Data file. $n = 3$ independent experiments (**c, f, g**). Data (**c, f, g**) are presented as mean values ± SEM. For gels (**d–e**) experiments were independently repeated twice with similar results.

*Lm*PC-PLC crystallized in space group P2$_1$2$_1$2$_1$ with two molecules, molA and molB, in the asymmetric unit (Fig. 2b), whose structures were determined at 2 Å resolution (Supplementary Table 1, Supplementary Fig. 5). Three metal ions were found bound to the active site in molA, and only two in molB (Fig. 2b). The positions of the metals ions were numbered according to the crystal structure of *Bc*PC-PLC[24]. To show that the metal ions bound to active site of *Lm*PC-PLC were Zn ions, we also collected data at a wavelength of 1.33 Å, where Zn should not contribute to the anomalous signal. For the position of the metal ion 1, we observed a signal in the anomalous difference map of the data collected at 1.26 Å

X-ray wavelength, which disappeared in the anomalous difference map of 1.33 Å data (Supplementary Fig. 6a, b). This suggested that the metal ion at the position 1 was indeed Zn. On the other hand, anomalous difference map from the data collected at 1.33 Å showed strong peaks at metal ion position 2 and especially at position 3, indicating the presence of metal ions other than Zn (Supplementary Fig. 6a, b). Inductively coupled plasma optical emission spectroscopy (ICP-OES) and inductively coupled plasma mass spectrometry (ICP-MS) analyzes showed that Fe ions are present in the *Lm*PC-PLC sample in a significantly greater proportion in comparison to Zn ions, namely 1.43 Fe and 0.01 Zn per

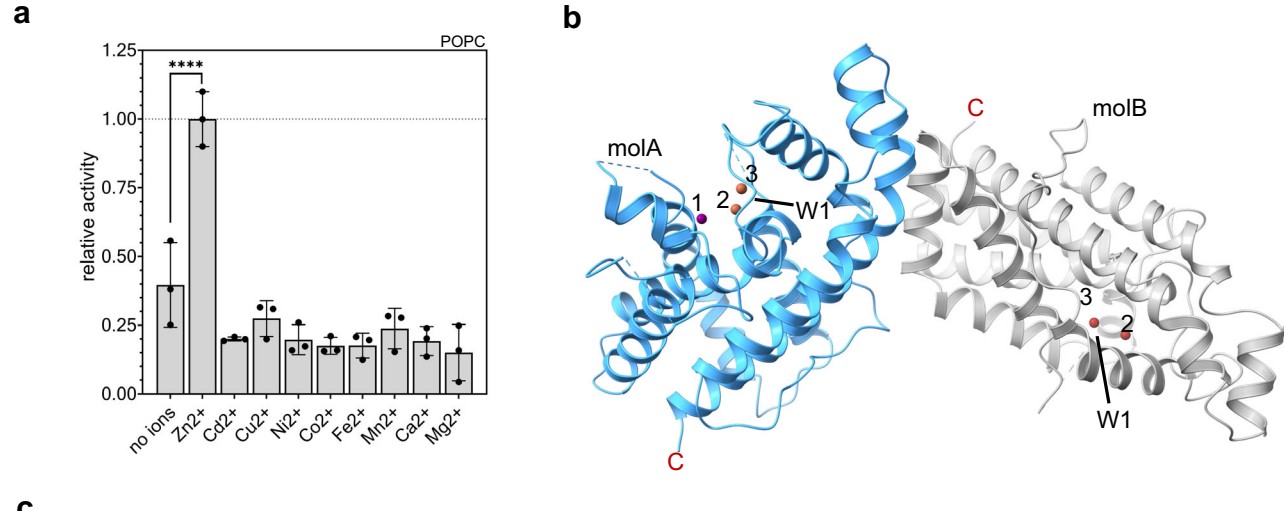

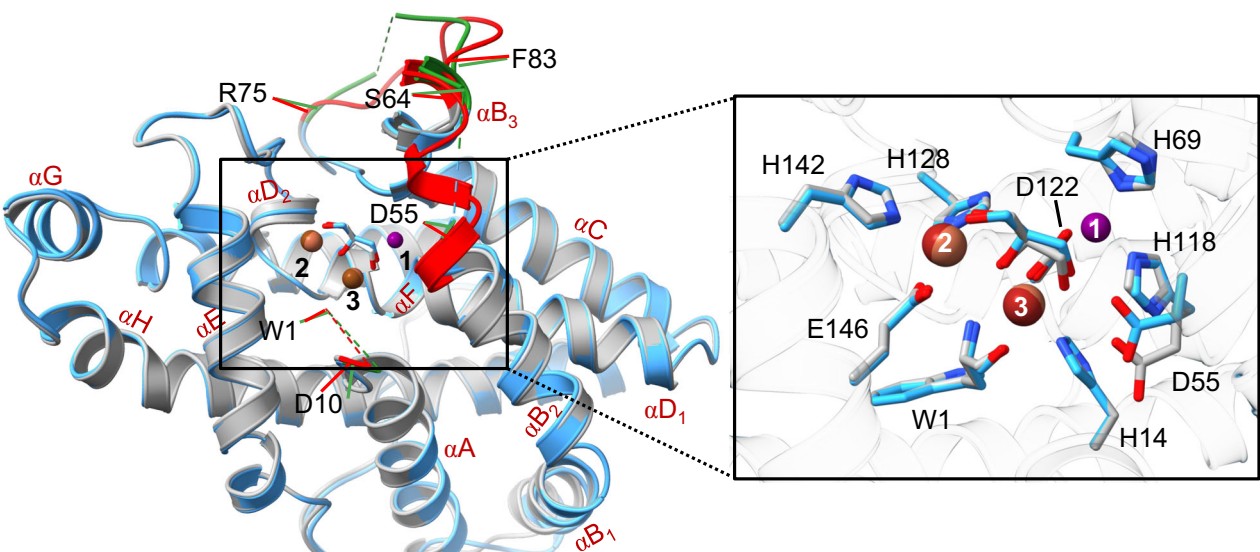

**Fig. 2 | Crystal structure of Zn²⁺-dependent *Lm*PC-PLC. a** Activity of WT *Lm*PC-PLC (50 nM) towards 100% POPC MLVs (4.5 mM) in the presence of various bivalent cations (50 µM) in 20 mM MES pH 6.5, 150 mM NaCl. Dunnett's multiple comparisons test was performed, ns: $P > 0.05$, *$P < 0.05$, **$P < 0.01$, ***$P < 0.001$, ****$P < 0.0001$. $n = 3$ independent experiments. Data are presented as mean values ± SEM. Source data are provided as a Source Data file. **b** Content of the asymmetric unit with marked protein termini (W1 for N-terminus and C as C-terminus) and Zn and Fe ions in the active site (violet and brown spheres,

respectively). MolA of *Lm*PC-PLC is shown in blue ribbon and molB in gray. **c** Left: Superposition of *Lm*PC-PLC molA (blue) and molB (gray). The loops that differ between the molA and mol B are marked in green and red, respectively. The loop S2-T9 is not defined in either of the molecules. α-helices are marked with letters A–H, based on the structure of *Bc*PC-PLC[24]. Right: zoom in into the active sites of molA and molB. Zn and Fe ion binding residues are shown in sticks. A glycerol molecule is bound in the active site of both molecules and is shown in sticks. Zn and Fe ions are shown as violet and brown spheres, respectively.

*Lm*PC-PLC molecule (Supplementary Fig. 6c). Interestingly, while the strength of the anomalous signal at the metal ion position 3 was comparable between the anomalous difference maps of 1.26 Å and 1.33 Å data, the anomalous signal at metal ion position 2 was significantly weaker in the anomalous difference map of the data collected at 1.33 Å (Supplementary Fig. 6a, b), suggesting that the metal ion position 2 may contain Fe ions in some molecules and Zn ions in others. It should be noted that no Zn or Fe ions were added during *Lm*PC-PLC purification or crystallization. The absence of a pronounced electron density in the anomalous difference map in molB at metal position 1 (Supplementary Fig. 6a) and the shorter distance between H69 and H118 than in molA indicated that no Zn or other metal ion was present at this position in molB (Fig. 2c). The absence of Zn1 in molB and the lower occupancy of Zn1 in molA compared with Fe2 and Fe3 (Supplementary Table 2) is

consistent with a previous report of the lower affinity of one of the Zn ions in *Lm*PC-PLC[26]. A glycerol molecule was bound in the active site of both molecules (Fig. 2c), originating from either the bacterial growth medium or the cell lysis buffer. Its three hydroxyl groups occupy similar positions to those of the water molecules in the crystal structure of *Bc*PC-PLC[24], with the Zn ion having tetrahedral coordination and both Fe ions having octahedral coordination (Supplementary Fig. 7).

## The crystal structure of *Lm*PC-PLC reveals a conserved PC-PLC folding with unique details

The structures of *Lm*PC-PLC molA and molB are very similar with an overall RMSD of 0.55 Å[44]. They are globular and consist of 11 α-helices, as found for the homologs from *Bc* and *Cp*[24,25] (Fig. 3a). The overall RMSD value between Cα atoms of *Lm*PC-PLC molB and *Bc*PC-PLC

(PDB-ID 1AH7, https://doi.org/10.2210/pdb1AH7/pdb) is 1.19 Å, and 2.09 Å with the N-terminal domain of *Cp*PC-PLC (PDB-ID 1CA1, https://doi.org/10.2210/pdb1CA1/pdb)[44]. Interestingly, the N-terminal residue W1 is well defined by electron density in both *Lm*PC-PLC structures (Supplementary Fig. 5a), while there was no traceable density for the following residues S2 to T9 connecting W1 to the αA-helix. This is likely due to flexibility, as the presence of these residues was confirmed by N-terminal sequencing (Fig. 1b). This is in contrast to *Bc*- or *Cp*PC-PLC structures[24,25], where the loop between the conserved W1 and αA-helix is structured and forms part of the active site wall (Supplementary Fig. 8). In addition, residues K57-Y61 and N77-L80 could not be built in molA due to the poor electron density, whereas they could be traced in molB (Fig. 2c). These residues are located in two loops, D55-S64 and R75-F83, respectively, which adopt different conformations in molA and molB (Fig. 2c). The orientation of molA with respect to molB in the asymmetric unit, with only 4.4% of the surface area[45] of each molecule involved in the interaction (Fig. 2b), suggests that only the monomeric form of *Lm*PC-PLC crystallized.

### The active site cleft of *Lm*PC-PLC is surrounded by flexible loops

The flexible loops S2-T9, D55-S64 and R75-F83 in *Lm*PC-PLC surround approximately half of the active site cleft (Fig. 2c). This is in contrast to *Bc*PC-PLC (Supplementary Fig. 8a), where the architecture of the active site cleft is defined and robust, as it does not change upon substrate binding[24,28]. On the other hand, the active site of *Cp*PC-PLC can adopt two states, open-active[25] and closed-inactive[29]. Two flexible loops, P55-T93 and D132-A146, are sandwiched between the PLC and C2 domains of *Cp*PC-PLC and adopt distinctly different conformations between the two structures (Supplementary Fig. 8b). The loss of activity in the closed conformation of *Cp*PC-PLC is achieved by the blocking of the active site by the two loops as well as the inability to bind the third Zn ion[29]. Similarly, the conformation of the active site cleft of *Lm*PC-PLC may switch between the states. *Lm*PC-PLC molB, with its partially occluded active site by the D55-S64 loop and the absence of Zn1, could represent the closed, inactive state, and molA with a more open active site due to the flexibility of the D55-S64 loop and Zn ion in place, could represent the open, active state (Fig. 2c). Moreover, similarly to the two bacterial homologs, *Lm*PC-PLC also contains several surfaces exposed tyrosine and phenylalanine residues, many of which are conserved (Supplementary Fig. 1, Supplementary Fig. 8c, d). Interestingly, most of them are concentrated on one side of the molecule, also around the active site cleft. This especially true for the region including three surface exposed Tyr residues in

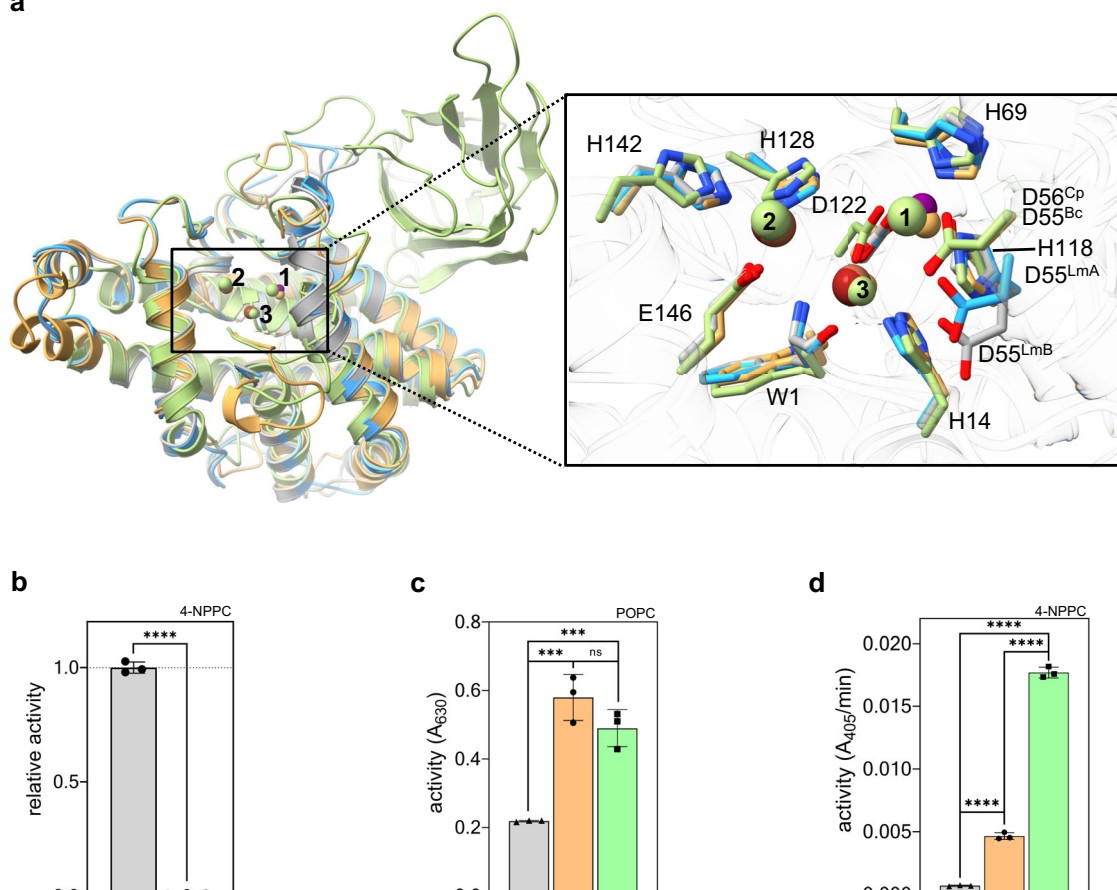

**Fig. 3 | Structural and functional comparison of *Lm*PC-PLC with the bacterial homologs. a** Left: Superposition of all three bacterial PC-PLC homologs (*Bc*PC-PLC: PDB-ID 1 AH7, orange ribbons; *Cp*PC-PLC: PDB-ID 1CA1, green ribbons; *Lm*PC-PLC molB, gray ribbons). Right: Zoom in into the active sites of the three homologs. Zn ions are shown as spheres in the same colors as ribbons in *Bc*- and *Cp*PC-PLC, except for violet for Zn and brown for Fe in *Lm*PC-PLC, and numbered. Zn/Fe ion coordinating residues are shown in sticks and labeled. In *Cp*PC-PLC, metal ion positions 1 and 2 are occupied by Cd ions that were present in the crystallization buffer. **b** Enzymatic activity of *Lm*PC-PLC WT and D55N mutant (500 nM) towards 4-NPPC (1 mM). **c** Activity of PC-PLC homologs (50 nM) towards 100% POPC MLVs (4.5 mM). **d** Activity of PC-PLC homologs (500 nM) towards 4-NPPC (1 mM). Experiments in b, c and d were performed in 20 mM MES pH 6.5, 150 mM NaCl, 50 (**c**) or 500 (**b**, **d**) μM ZnSO₄, and 1 mM CaCl₂ (only for *Cp*) at 37 °C. Dunnett's multiple comparisons test (**c**, **d**) or Student's t-test (**b**) was performed, ns: $P > 0.05$, *$P < 0.05$, **$P < 0.01$, ***$P < 0.001$, ****$P < 0.0001$. Source data are provided as a Source Data file. $n = 3$ independent experiments (**b**–**d**). Data in (**b**–**d**) are presented as mean values ± SEM.

the dynamic loops D55-S64 and R75-F83 (Y60, Y61, Y79) that could be involved in membrane binding and thus the stabilization of the dynamic loops.

## Structural plasticity of the active site and oligomerization regulate the enzymatic activity of *Lm*PC-PLC

D55 is located at the beginning of the flexible loop D55-S64 in *Lm*PC-PLC (Fig. 2c). This residue is conserved among bacterial homologs (Supplementary Fig. 1) and coordinates Zn1 in *Bc*PC-PLC[24] and *Cp*PC-PLC[25] (Fig. 3a). The position of D55 is different in molA and molB of *Lm*PC-PLC (Fig. 2c), and it is in both molecules spatially displaced compared to the corresponding D55 and D56 in *Bc*PC-PLC and *Cp*PC-PLC, respectively (Fig. 3a). In molA, the side chain of D55 is too far from Zn1 to coordinate, and in the absence of Zn1 in molB, D55 is even further away (Fig. 2c). This could explain the lower affinity for Zn ions in *Lm*PC-PLC[26] and a low occupancy of the Zn1 site in *Lm*PC-PLC molA and the absence of Zn1 in molB (Supplementary Fig. 6, Supplementary Table 2).

Interestingly, the enzymatic activity of *Lm*PC-PLC was significantly lower than that of the two homologs, regardless of pH or substrate (Supplementary Fig. 9a, b). Further comparative enzymatic assays between the homologs were performed at pH 6.5, as this was their optimal pH in our system. All three PC-PLCs were largely activated by the addition of $Zn^{2+}$, and in the case of *Cp*PC-PLC, also $Ca^{2+}$, and inhibited by the metal chelating agent O-phenanthroline[46] (O-phe) (Supplementary Fig. 9c). Another metal chelator, N,N,N′,N′-tetrakis(2-pyridinylmethyl)−1,2-ethanediamine (TPEN), reported to have high affinity for Zn ions[47] showed even stronger inhibition of *Lm*PC-PLC than O-phe (Supplementary Fig. 10). *Lm*PC-PLC showed 2-3-fold lower activity for POPC, and 7- and 25-fold lower activity for the synthetic substrate at pH 6.5 compared to *Bc*PC-PLC and *Cp*PC-PLC, respectively (Fig. 3c, d). However, despite the spatial dislocation of D55, this residue is still crucial for the enzymatic activity of *Lm*PC-PLC, as the *Lm*PC-PLC[D55N] mutant was inactive (Fig. 3b)[26] and could play a similar role in catalysis as its counterpart, D55 in *Bc*PC-PLC, where it was suggested to play the role of a general base via a nucleophilic attack on a water molecule bound to Zn1 and Zn2[28,48,49].

## The exact positioning of W1 is crucial for the enzymatic activity of *Lm*PC-PLC

The conserved N-terminal residue W1, is firmly anchored in the active site cleft (Figs. 2c and 4a). Its main chain atoms coordinate Fe3, while the NE1 atom from the side chain anchors it deep in the active site by forming a hydrogen bond with the S124 side chain and the main chain carbonyl of T121 (Fig. 4a). The positioning of the indole ring of W1 is further supported by the pocket lined by hydrophobic residues L17, Y145, V149, and L218. We replaced W1 with A, E, F or K and removed W1 and S2 (ΔWS), as removal of only W1 was not possible in the intein-fusion system[50]. These mutations did not affect protein folding (Supplementary Fig. 4b), and all showed oligomer formation on SEC (Supplementary Fig. 11). However, all W1 substitutions except W1F significantly decreased the activity of *Lm*PC-PLC for the lipid and the synthetic substrate (Fig. 4b, c).

These results indicated that the precise positioning of all atoms of the N-terminal residue (Fig. 4a) is required for the formation of the active site cleft of *Lm*PC-PLC, which could lead to productive catalysis. It is therefore not surprising that already the addition of a residue at the N-terminus of W1 turns off the enzymatic activity of *Lm*PC-PLC[51], since the placement of a new residue would require a change in the position of W1.

## The propeptide added in trans specifically inhibits *Lm*PC-PLC

*Lm*PC-PLC is synthesized as an inactive proenzyme whose N-terminal propeptide contains 26 amino acids[14]. Additional amino acids N-terminal to W1 may contribute to protein stability and functional regulation required in various bacterial compartments during protein maturation. The propeptide does not appear to be needed for proper folding of the mature enzyme, as it has been shown here and elsewhere[26,52] that the active *Lm*PC-PLC can be readily produced in its absence.

We attempted to determine the crystal structure of pro-*Lm*PC-PLC. pro-*Lm*PC-PLC was successfully purified and although it eluted from the SEC column in one peak, it showed a tendency to oligomerize at higher concentrations (Supplementary Fig. 12a–d). The CD spectrum of pro-*Lm*PC-PLC was similar to that of the mature enzyme (Supplementary Fig. 12e), indicating the mature part of the protein is already formed in the proform. As expected, pro-*Lm*PC-PLC was enzymatically inactive (Supplementary Fig. 12f). However, pro-*Lm*PC-PLC was prone to proteolysis and aggregation, thereby preventing crystallization.

We then used AlphaFold2[53] (AF2), which suggested the position of the propeptide on the surface of the mature *Lm*PC-PLC with its N-terminus directly above the active site (Supplementary Fig. 13a). Although the predicted model of the pro-*Lm*PC-PLC structure has a relatively low per-residue confidence score pLDDT in the propeptide region (Supplementary Fig. 13b), we found the predicted model interesting because it suggested that the second of the two cysteines forms a disulfide bond with C143 on the surface of the mature enzyme (Supplementary Fig. 13a). On this basis, we designed two peptides, the full-length propeptide containing 26 residues, NS-26, and a peptide containing six N-terminal residues of the propeptide, NE-6 (Fig. 4d), which we further used in inhibition experiments.

Both peptides inhibited the enzymatic activity of *Lm*PC-PLC toward the synthetic substrate 4-NPPC, but NS-26 was significantly stronger than NE-6, with IC50 values of 9.4 μM (95% CI 4.9 to 33.9 μM) and 43.8 μM (95% CI 36.1 to 54.7 μM), respectively (Fig. 4d). NE-6 is generally a weaker inhibitor than NS-26, for both substrates (Fig. 4d-h). This suggests that the longer peptide NS-26 is required for more efficient inhibition of *Lm*PC-PLC, likely due to more efficient occlusion of the active site cleft in the case of the longer peptide NS-26. In contrast to *Lm*PC-PLC, inhibition of *Bc*- and *Cp*PC-PLC was much less effective in the case of the small substrate 4-NPPC (Fig. 4e), and was not even detected in the case of the lipid substrate (Fig. 4f), indicating that inhibition by the two peptides was specific to *Lm*PC-PLC.

The two cysteines in the peptides could possibly interact with the two surface-exposed cysteine residues of the mature part of *Lm*PC-PLC (Supplementary Fig. 13a). The reducing agent TCEP could not be used in our assays because it interfered with the assay, probably by depleting Zn ions (Supplementary Fig. 13c). To circumvent this, we examined the effect of the peptides on the cysteine mutants *Lm*PC-PLC[C143S], *Lm*PC-PLC[C168S] and *Lm*PC-PLC[C143S+C168S]. The inhibition of *Lm*PC-PLC[C168S] followed the trend observed for wild-type *Lm*PC-PLC, whereas the inhibition of *Lm*PC-PLC[C143S] or *Lm*PC-PLC[C143S+C168S] was weaker, especially for 4-NPPC as substrate (Fig. 4g, h). This suggests that C143, located near the active site, may be involved in (pro)peptide-induced inhibition, as also indicated by the AF2 model of pro-*Lm*PC-PLC (Supplementary Fig. 13a). However, the C143S mutation does not completely prevent inhibition of *Lm*PC-PLC[C143S] or *Lm*PC-PLC[C143S+C168S] by either peptide, suggesting the role of other residues of the active site cleft and the propeptide in successful inhibition of the enzymatic activity of *Lm*PC-PLC.

## Phospholipase activity of *Lm*PC-PLC facilitates pore-formation by LLO

During the infection cycle, *Lm*PC-PLC acts together with the pore-forming toxin LLO to disintegrate the vacuolar membrane[12]. To test their interplay at the molecular level, we first investigated how *Lm*PC-PLC affects the first step of LLO action, the binding to lipid membranes. We preincubated MLVs composed of POPC and CHOL with different lipid ratios with *Lm*PC-PLC, then added LLO and analyzed the

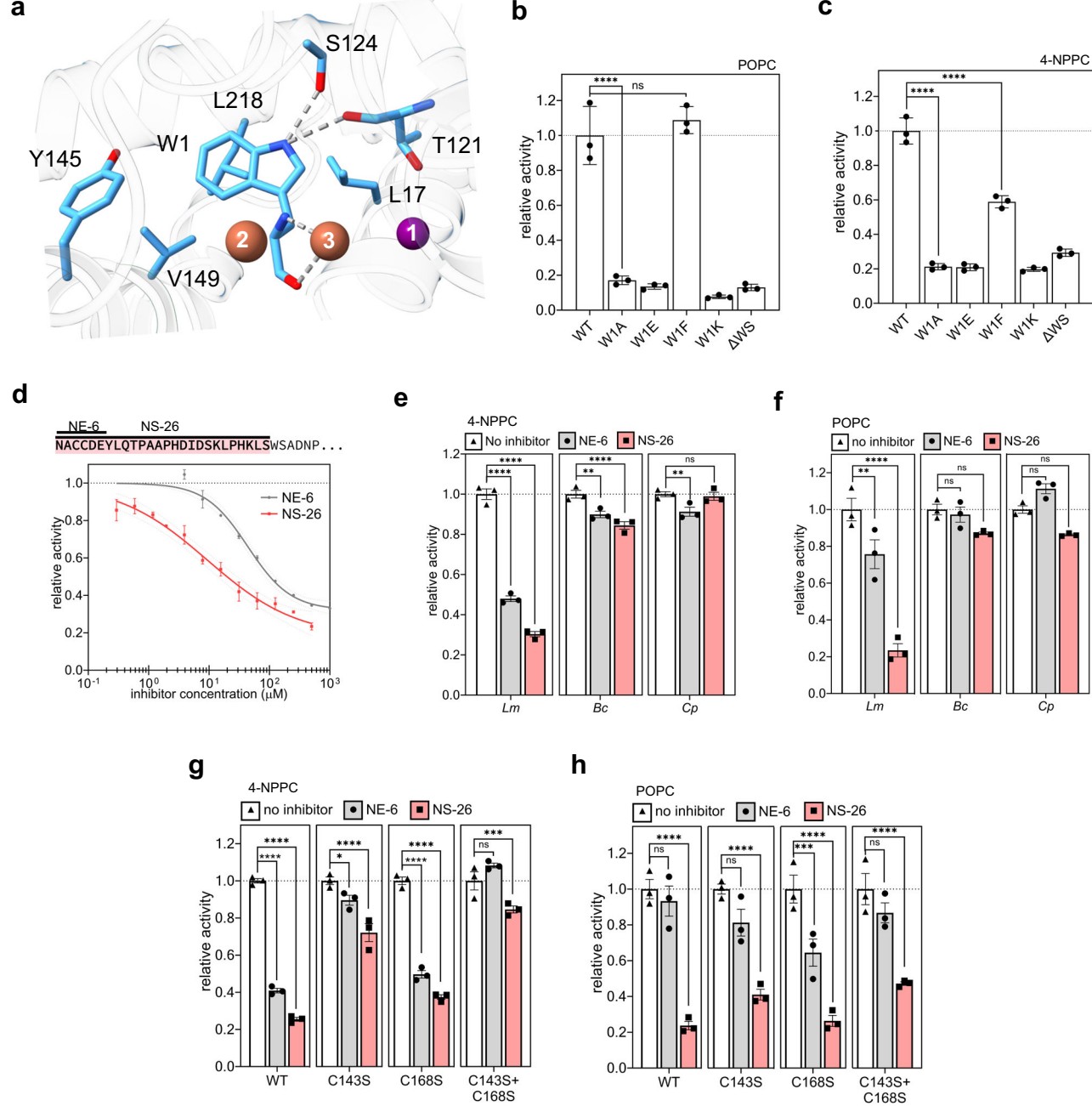

**Fig. 4 | Regulation of the enzymatic activity of *Lm*PC-PLC by the N-terminal W1 and the propeptide added in trans. a** W1 (sticks) in the active site of *Lm*PC-PLC, molB. Interactions between W1 and other residues and Fe3 are shown as dotted gray lines. Zn (position 1) and Fe (positions 2 and 3) ions: numbered violet (Zn) and brown (Fe) spheres. **b** Activity of the W1 mutants (50 nM) relative to WT *Lm*PC-PLC towards 100% POPC MLVs (4.5 mM). **c** Activity of W1 mutants (500 nM) relative to WT *Lm*PC-PLC towards 4-NPPC (1 mM). **d** Top: Amino-acid sequence of the *Lm*PC-PLC propeptide (highlighted in pink), and the synthetic peptides NE6 and NS-26 used in assays. Bottom: Dose-dependent inhibition of *Lm*PC-PLC (500 nM) by the two peptides towards 4-NPPC (1 mM). **e** Inhibition of activity of PC-PLC homologs (500 nM) by the two peptides (100 µM) on 4-NPPC (1 mM). **f** Inhibition of activity of

PC-PLC homologs (50 nM) by the two peptides (10 µM), on 100% POPC MLVs (4.5 mM). **g** Inhibition of *Lm*PC-PLC cysteine mutants (500 nM) towards 4-NPPC (1 mM) by the two peptides (100 µM) in comparison with WT *Lm*PC-PLC (500 nM). **h** Inhibition of *Lm*PC-PLC cysteine mutants (50 nM) on 100% POPC MLVs (4.5 mM) by the peptides (10 µM) in comparison to the WT *Lm*PC-PLC (50 nM). Experiments were performed in 20 mM MES pH 6.5, 150 mM NaCl, 50 (**b**, **f**, **h**) or 500 µM (**c**, **d**, **e**, **g**) ZnSO$_4$, and 1 mM CaCl$_2$ (only for *Cp*) at 37 °C. Dunnett's multiple comparisons test was performed (**b**, **c**, **e–h**), ns: $P > 0.05$, *$P < 0.05$, **$P < 0.01$, ***$P < 0.001$, ****$P < 0.0001$. $n = 3$ independent experiments (**b–h**). Data (**b–h**) are presented as mean values ± SEM. Source data are provided as a Source Data file.

binding of LLO to liposomes using the co-sedimentation assay (Fig. 5a). LLO effectively bound to MLVs with 50 mol% CHOL, regardless of preincubation with *Lm*PC-PLC. On the other hand, the preincubation of vesicles with *Lm*PC-PLC more than doubled the binding of LLO to membranes with 10 mol%, 20 mol% or 30 mol% CHOL (Fig. 5a, Supplementary Fig. 14a).

We then tested how *Lm*PC-PLC affects the permeabilization of the membrane by LLO. As a model system, we used large unilamellar vesicles (LUVs) with different POPC:CHOL molar ratios containing a self-quenching fluorescent dye 5(6)-Carboxyfluorescein (CF). At 40 mol% CHOL, LLO perforated the vesicles as expected, regardless of preincubation with *Lm*PC-PLC (Fig. 5b). Interestingly, at 40 mol%

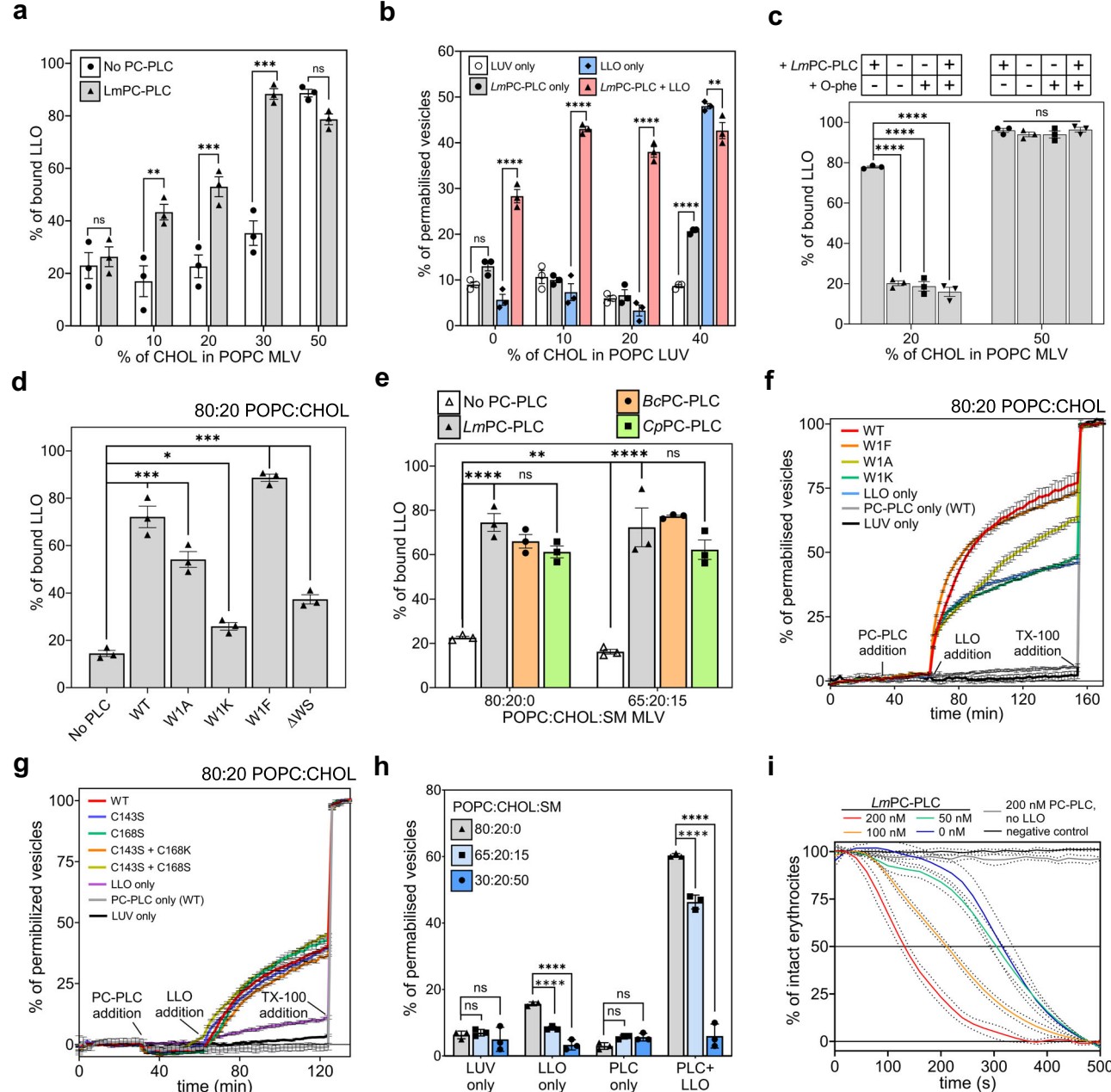

**Fig. 5 | Mature *Lm*PC-PLC promotes LLO binding to lipid membranes and transmembrane pore-formation. a** Binding of LLO (3 µM) to POPC:CHOL MLVs (8 mM), with or without preincubation of MLVs with *Lm*PC-PLC (1.4 µM). **b** Vesicle permeabilization assay with LLO (0.5 µM) of 200 nm POPC:CHOL LUVs with, or without preincubation of LUVs by *Lm*PC-PLC (2 µM). **c** Binding of LLO (3 µM) to POPC:CHOL MLVs (8 mM), with or without *Lm*PC-PLC (1.4 µM) preincubation, in the absence or presence O-phenanthroline (O-phe, 1 mM). **d** Binding of LLO (3 µM) to POPC:CHOL MLVs (8 mM), with or without preincubation with *Lm*PC-PLC W1 mutants (1.4 µM). **e** Binding of LLO (3 µM) to POPC:CHOL:SM MLVs (8 mM) with or without preincubation with PC-PLC homologs (1.4 µM). **f** Kinetics of vesicle permeabilization assay of 200 nm LUVs with LLO (0.5 µM), with or without preincubation of LUVs by *Lm*PC-PLC WT or W1 mutants (2 µM). **g** Kinetics of vesicle permeabilization assay of 200 nm LUVs with LLO (0.5 µM) with or without preincubation of LUVs by *Lm*PC-PLC WT or cysteine mutants (2 µM). **h** Vesicle

permeabilization assay of 200 nm POPC:CHOL:SM LUVs with LLO (0.5 µM) with or without *Lm*PC-PLC preincubation (2 µM). **i** Hemolytic assay using bovine erythrocytes (OD$_{600}$ = 0.5) of LLO (0.25 nM) in the absence or presence of *Lm*PC-PLC (50, 100 or 200 nM). All wells with LLO reached complete lysis of erythrocytes within 500 seconds. Negative control: erythrocytes only. Experiments were performed in 20 mM MES pH 6.5, 150 mM NaCl, 50 µM ZnSO$_4$, and 1 mM CaCl$_2$ (only for *Cp*) at room temperature (**a, c, d, e**) or 37 °C (**b, f, g, h, i**). LUV only represents the experiment with no added proteins. All experiments were done in triplicates. Student's t-test (**a, b**) or Dunnett's multiple comparisons test (**c, d, e, h**) were performed, ns: $P > 0.05$, $*P < 0.05$, $**P < 0.01$, $***P < 0.001$, $****P < 0.0001$. Lipid content of vesicles is presented as molar% or molar ratio. Source data are provided as a Source Data file. $n = 3$ independent experiments (**a–i**). Data (**a–i**) are presented as mean values ± SEM.

CHOL, significant release of CF was observed even with *Lm*PC-PLC alone (Fig. 5b). This could be due to damage of the vesicles due to increased phospholipase activity at high concentrations of CHOL (Supplementary Fig. 14b). Importantly, LLO alone was unable to

permeabilize LUVs at 0 mol%, 10 mol% or 20 mol% CHOL, whereas it successfully permeabilized them when preincubated with *Lm*PC-PLC, achieving values similar to those obtained at 40 mol% of CHOL (Fig. 5b). Interestingly, a considerable increase in the permeabilization

of the vesicles by LLO was observed for 100 mol% POPC LUVs after preincubation with *Lm*PC-PLC, possibly due to the ability of LLO to bind to membrane defects or membrane edges[9].

We then measured the binding of LLO to POPC:CHOL MLVs containing 20 mol% CHOL, which were preincubated either with *Lm*PC-PLC and its inhibitor O-phe (Fig. 5c, Supplementary Fig. 15a), *Lm*PC-PLC W1 mutants (Fig. 5d, Supplementary Fig. 15b), cysteine mutants of *Lm*PC-PLC (Supplementary Fig. 16) or *Bc* and *Cp* PC-PLC homologs (Fig. 5e, Supplementary Fig. 17a). These results showed that the increase in LLO binding to membranes was correlated with the enzymatic activity of *Lm*PC-PLC and that the phospholipase activity required for increased LLO binding was not bacterial species specific. Furthermore, we measured CF-release from POPC:CHOL LUVs containing 20 mol% CHOL, by LLO from vesicles preincubated with *Lm*PC-PLC W1 mutants and single and double cysteine mutants (Fig. 5f, g). These results further confirmed that phospholipase activity facilitates not only LLO binding, but also permeabilization of the vesicles by LLO.

Next, we increased the complexity of the membranes and prepared vesicles consisting of a three-component membrane POPC:CHOL:SM in the molar ratio 65:20:15. SM is ubiquitous in the animal plasma membrane, which typically contains about 15 mol% of this lipid[54]. Therefore, we have included SM in our lipid membrane models. When the MLVs were preincubated with *Lm*PC-PLC, the binding of LLO to the POPC:CHOL:SM membrane was increased approximately four-fold and reached a similar level to the POPC:CHOL membrane (Fig. 5e). As expected, binding of LLO to membranes containing SM was stimulated to similar levels by *Bc*PC-PLC and *Cp*PC-PLC (Fig. 5e), as both are enzymatically active on three-component vesicles (Supplementary Fig. 17b). Interestingly, although the presence of SM in lipid membranes reduces the enzymatic activity of *Bc*PC-PLC, the residual activity of *Bc*PC-PLC still strongly stimulates the binding of LLO to the SM-containing membrane.

The synergy between *Lm*PC-PLC and membrane permeabilization by LLO was also observed on POPC:CHOL:SM vesicles. We prepared LUVs containing POPC, 20 mol% CHOL and SM at 0 mol%, 15 mol%, or 50 mol%. At 0 or 15 mol% SM, preincubation of the vesicles with *Lm*PC-PLC resulted in a dramatically increased release of CF compared to permeabilization by LLO alone (Fig. 5h). Importantly, the introduction of SM into the membranes decreased the activity of LLO, which was less affected at 15 mol% of SM than at 50 mol% SM, where no LLO-induced permeabilization was observed even with *Lm*PC-PLC pre-treatment (Fig. 5h). This could be due to the lower availability of CHOL with increasing concentrations of SM in the membranes[55].

Finally, we performed the hemolysis assay. Although the membranes of erythrocytes are not the natural target of LLO, they are used as a model of a natural complex membrane, because their plasma membrane contains a high CHOL concentration (ca. 50 mol%[56]), which is more than other cell membranes in the human body (20-50 mol%)[54]. Despite high the CHOL content in the erythrocyte membranes, which made LLO highly hemolytic on its own, the hemolysis rate was nevertheless significantly increased in the presence of *Lm*PC-PLC (Fig. 5i). Furthermore, the rate of hemolysis by LLO was dependent on the concentration of *Lm*PC-PLC, further supporting the role of active phospholipase in this process.

### *Lm*PC-PLC influences the morphology of LLO oligomers on lipid vesicles

We used cryo-transmission electron microscopy (cryo-EM) to investigate how the positive synergy between *Lm*PC-PLC and LLO affects the mode of membrane disruption by LLO. We first followed the pore formation by LLO in POPC:CHOL (molar ratio 1:1) LUVs at different times and temperatures (Supplementary Fig. 18). These experiments showed prevailing formation of LLO arcs at early stages, with slit- and ring-shape pores predominating with time. The transmembrane pores were of different sizes and shape, suggesting merging between the

arcs or arcs and rounded pores. The process of pore formation was expectedly faster at a higher temperature.

Next, we imaged vesicles of different lipid compositions in the presence of *Lm*PC-PLC alone, as binding of *Lm*PC-PLC to virtually all types of membranes could be detected (Supplementary Figs. 14-17). Under the chosen conditions, no extensive binding of *Lm*PC-PLC to membranes could be observed by cryo-EM (Supplementary Fig. 19). However, we detected individual vesicles with changes in the membrane bilayer in a form of rough patches that could be attributed to either binding of *Lm*PC-PLC or lipid hydrolysis (Supplementary Fig. 19, right column).

To observe the synergy of LLO and *Lm*PC-PLC, we prepared POPC:CHOL LUVs with different CHOL content at pH 6.5. When exposed to LLO alone, no membrane defects were observed on LUVs made of 100% POPC or POPC:CHOL LUVs with 20 mol% CHOL, while a small number of arcs, slits and pores were observed at 30 mol% CHOL, and, as expected, even more at 50 mol% CHOL (Fig. 6). When 100% POPC LUVs were preincubated with *Lm*PC-PLC prior to addition of LLO, a small portion of the vesicles appeared to be covered by protein (Fig. 6). Preincubation of LUVs containing 20–50 mol% CHOL with *Lm*PC-PLC, followed by the addition of LLO, resulted in the formation of the sea urchin-like clusters, indicating the presence of membrane-bound elongated LLO molecules[21]. In addition, we also observed disrupted lipid membranes and isolated LLO arcs (Fig. 6, Supplementary Fig. 20). The morphology of these disrupted vesicles differed markedly from that of vesicles exposed to LLO alone (Fig. 6). When three-component lipid vesicles were used, POPC:CHOL:SM, molar ratio 65:20:15, LLO oligomers and sea urchin-shaped disrupted vesicles were observed only when LUVs were preincubated with *Lm*PC-PLC (Fig. 6), which is consistent with the CF-release experiment (Fig. 5b).

In summary, for each lipid composition of LUVs used, we found a clear difference between vesicles perforated with LLO alone or in combination with *Lm*PC-PLC. When LLO was used alone, we observed mainly membrane bound LLO oligomers in the shape of arcs, as well as membrane inserted slits and round pores. In contrast, such structures were hardly observed when LUVs were preincubated with *Lm*PC-PLC. In the presence of *Lm*PC-PLC, membranes disruption by LLO appeared to be much more aggressive, resulting in more pronounced disintegration of membranes with LLO arcs floating in solution (Supplementary Fig. 20b) and protein-membrane patches in the shape of a sea-urchin. Probably due to the higher availability of free CHOL facilitated by the phospholipase activity of *Lm*PC-PLC, a considerable amount of LLO appears to remain on the membrane in an uninserted form. Consequently, there is not enough space for the arcs to grow or merge into round shaped pores, hence arcs as well as high load of bound proteins still effectively disturb the vesicle membranes.

## Discussion

Listeriosis is one of the deadliest foodborne diseases. The intracellular nature of the infection, the ability to cross the blood-brain barrier and the placental barrier, the high temperature, salt and pH stability of the bacterium, and the emergence of antibiotic-resistant and multidrug-resistant strains pose a significant health risk to humans and animals worldwide[57–59]. Therefore, the pathophysiology of *Lm* in the cellular environment, including the interplay of different virulence factors, has been studied in detail[5]. The central aim of our study was to contribute to the understanding of the mechanism of action of one of the most important virulence factors of *Lm*, broad-range phospholipase C, *Lm*PC-PLC, and its molecular interplay with the main virulence factor, the pore-forming toxin LLO, in the process of lipid membrane disintegration.

Our results can be summarized in six main points. First, we show that *Lm*PC-PLC tends to both oligomerize and aggregate, with different oligomeric states in equilibrium with each other. Moreover, oligomerization appears to largely attenuate the enzymatic activity

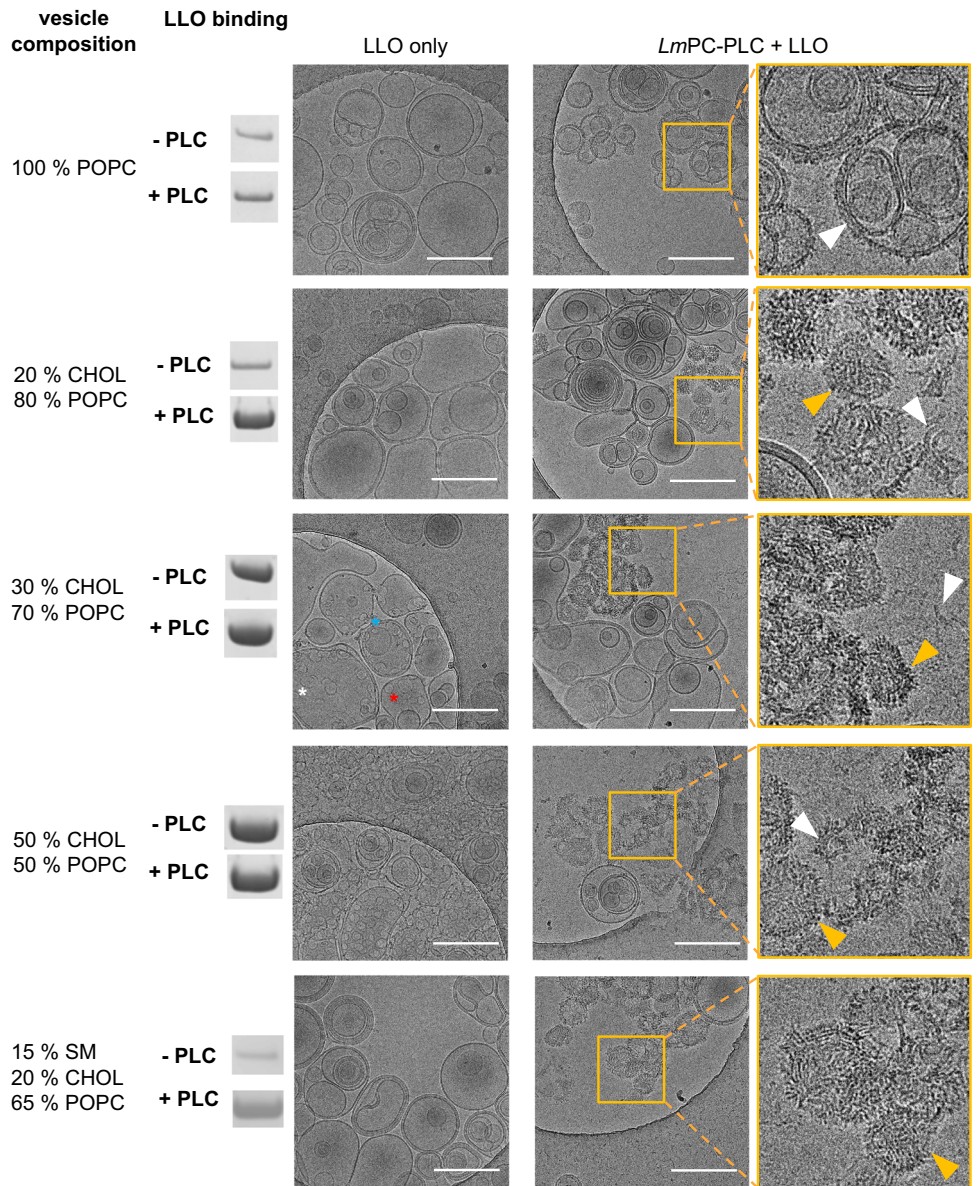

**Fig. 6 | *Lm*PC-PLC affects the mode of membrane disruption by LLO.** Left: composition of lipid membranes of LUVs used in analysis. SDS-PAGE gel bands shown are taken from Supplementary Figs. 14 and 17 and represent binding of LLO in the absence or presence of *Lm*PC-PLC. Right: Cryo-EM micrographs were obtained at 73,000x magnification with 200 nm LUVs (2.5 mM) consisting of POPC:CHOL:SM of various lipid ratios as indicated. Left column of micrographs (LLO only): LUVs exposed only to 5 μM LLO. White asterisk marks oligomeric arc on the membrane surface formed by LLO, red the oligomeric LLO pore, and blue asterisk marks LLO slit formed by two membrane inserted arcs. Middle column of micrographs: LUVs preincubated with 5 μM *Lm*PC-before addition of 5 μM LLO. Right column of micrographs: Close ups of the areas within yellow squares (250 nm x 250 nm). Orange arrows mark show LLO bound on the membrane. White arrows mark LLO arks extracted from the membrane. The size bar on micrographs is 250 nm. Buffer: 20 mM MES pH 6.5, 150 mM NaCl, 500 μM ZnSO₄. Three individual Cryo-EM experiments were performed, except for POPC:CHOL:SM membrane, where done twice, in all cases with similar results between repetitions.

of *Lm*PC-PLC, which may be one of the mechanisms regulating phospholipase activity in the host cell during the different steps of the infection cycle. The absence of oligomerization in homologous enzymes from extracellular bacteria *Bc* and *Cp* supports the possibility that oligomerization of *Lm*PC-PLC may be part of the regulatory mechanism related to the intracellular nature of *Lm*. Similarly, LLO is also largely unstable in the cytosol due to its structural properties. At neutral pH of the cytosol, LLO aggregates, which abolishes its membrane disrupting action inside the cytosol[60,61].

Second, we show that the phospholipase activity of *Lm*PC-PLC is significantly lower than that of the *Bc* and *Cp* homologs and is highly and specifically dependent on the addition of Zn ions. The crystal structure of *Lm*PC-PLC explained the strong dependence of this

enzyme on the addition of Zn ions (Fig. 2a) as well as the low affinity of the active site for the Zn ions[26]. Interestingly, in contrast to the *Bc*[24] and *Cp*[25] homologs and the predictions from previous studies on *Lm*PC-PLC[26], only one Zn ion (Zn1), rather than three, was found in the active site, and Fe ions were found at metal ion positions 2 and 3, i.e., Fe2 and Fe3, with the possibility that in some molecules Zn binds to metal position 2. Fe ions are bound to the active site with higher occupancy than Zn ions (Supplementary Table 2), with Zn1 even absent in a subpopulation of *Lm*PC-PLC molecules (molB). Our observation that the enzymatic activity of *Lm*PC-PLC is largely dependent only on Zn ions but not Fe ions, suggests that Fe ions may play a structural role in the recombinant *Lm*PC-PLC or possibly other metal ions present in *Lm*PC-PLC in its original environment. Although free Zn ions are scarce in the cell[62], *Lm* have evolved systems that facilitate the uptake of Zn ions

from the host[63] and virulence factors such as *Lm*PC-PLC may benefit from this.

Third, the active site of *Lm*PC-PLC is structurally plastic and its key catalytic residue, D55, is spatially shifted compared with the *Bc* and *Cp* homologs, where the corresponding residues D55 and D56, respectively, coordinate the Zn ion (Zn1) for which *Lm*PC-PLC has very low affinity. Therefore, the shifted position of D55 in *Lm*PC-PLC could be due to the low Zn1 occupancy and flexibility of the active site loops D55-S64, S2-T9, K57-Y61 and N77-L80. *Lm*PC-PLC may thus switch between an enzymatically inactive and an active state, triggered by binding of a lipid membrane substrate and saturation of the active site with Zn ions. In addition, based on the structure of *Bc*PC-PLC with the substrate analog, the choline binding pocket was identified. This consists of three choline binding residues, E4, Y56 and F66, with the latter two forming a π-cation interaction with the choline headgroup[64]. In *Lm*PC-PLC, only F66 is conserved in the primary structure, however, it occupies a different position in *Lm*PC-PLC than in *Bc*PC-PLC (Supplementary Fig. 8d). Furthermore, E4 is substituted with D4 in *Lm*PC-PLC and Y56 with H56. However, D4 in *Lm*PC-PLC is located in the flexible loop S2-T9, which is structurally not defined, and the position of H56 in *Lm*PC-PLC is offset in comparison to Y56 in *Bc*PC-PLC. Whereas the choline binding pocket appears to be formed in *Bc*PC-PLC already in the absence of the substrate, this does not seem to be the case for *Lm*PC-PLC. Previous studies have shown that D4/H56 in *Lm*PC-PLC could be replaced by *Bc*PC-PLC sequence with no harm to enzymatic activity[26,65]. Therefore, it is currently unclear if such binding pocket exists in *Lm*PC-PLC. Another possibility is that the pocket is formed upon membrane or substrate binding since *Lm*PC-PLC exhibits high flexibility in the regions that include those residues. The structural plasticity of the active site cleft in combination with the low affinity for Zn ions and the tendency to oligomerize into enzymatically much weaker oligomers could represent a regulatory mechanism the enzymatic activity of *Lm*PC-PLC.

Fourth, we have shown that *Lm*PC-PLC can be inhibited by addition of its propeptide in trans, and the inhibition is largely specific to *Lm*PC-PLC, as no significant inhibition was observed with *Bc*- or *Cp*PC-PLC. This is enabled by specific a sequence of both the propeptide and the mature part of the enzyme, which may include the disulfide bond formation between C143 in the mature part of *Lm*PC-PLC and one of the cysteines from the propeptide.

Fifth, our results clearly demonstrate a positive synergy between *Lm*PC-PLC and LLO at the molecular level during lipid membrane disintegration by LLO in vitro. This synergy is positively correlated with the enzymatic activity of *Lm*PC-PLC. We show that *Lm*PC-PLC facilitates membrane disruption by LLO in membranes where LLO alone cannot successfully act due to the limited amount of free CHOL receptor. This suggests that the phospholipase activity of *Lm*PC-PLC releases CHOL from complexes with other lipids in the membrane to become available as a receptor for LLO, as has been reported for the corresponding *Cp*PC-PLC/perfringolysin O pair of *Cp*[41]. Therefore, it was not surprising that we could observe similar effects on the pore-forming activity of LLO with *Bc*- and *Cp*-PC-PLC in our experiments. This result is consistent with a previous study[65] in which an *Lm* deletion mutant with a deleted gene encoding *Lm*PC-PLC was able to escape the vacuole when complemented with *Bc*PC-PLC. However, the bacterial viability, cell-to-cell spread and virulence in mice were significantly impaired, suggesting that *Bc*PC-PLC cannot fully complement *Lm*PC-PLC[65]. This is further evidence that LLO requires its cognate PC-PLC, which is adapted to intracellular life, for optimal performance.

Finally, in demonstrating the influence of *Lm*PC-PLC on membrane perforation by LLO, we found that complementation of the two proteins not only lowers the threshold concentration of CHOL for successful pore formation by LLO, but also alters morphology of LLO oligomers on lipid vesicles. However, this may also depend on other environmental factors[66] such as pH, temperature, the presence of other biologically relevant molecules, membrane composition, and the timing of the interaction, which warrants further investigation. In addition, previous studies have shown that *Lm*PC-PLC and LLO can act antagonistically under certain conditions during infection, i. e., in autophagy, reactive oxygen species production, calcium flux, mitochondrial damage and apoptosis[42,67,68]. Some of this antagonism might be caused by *Lm*PC-PLC mediated phosphocholine production inhibiting excessive LLO-induced damage in the cytosol[42]. Thus, the interplay between listerial virulence factors could be spatially and temporally regulated to ensure successful progression of the infection cycle. Together with our results, this demonstrates the nuanced and multifaceted interaction of the two virulence factors in the cytosolic niche.

## Methods

### Molecular cloning
The gene fragments transcribing the mature form of the wild-type *Lm*PC-PLC (UniProt ID P33378), LLO (UniProt ID P13128) and *Cp*PC-PLC (UniProt ID Q0TV31) were prepared by using genomic DNA isolates kindly provided by the Faculty of Veterinary Medicine at the University of Ljubljana, Slovenia. The gene encoding the mature wild-type *Bc*PC-PLC (UniProt ID P09598) and pro*Lm*PC-PLC (UniProt ID P33378) were purchased from Invitrogen (USA). Individual PC-PLC gene fragments were inserted into plasmid pTYB21 (NEB, USA), downstream and in-frame of the intein tag with chitin binding domain. LLO gene was inserted in pPROEX HTb vector downstream of the His6-tag and TEV recognition site. All *Lm*PC-PLC mutants were prepared using site directed mutagenesis by PCR[69]. All constructs were verified by nucleotide sequencing (Eurofins Genomics, Luxembourg). List of oligonucleotides used in cloning and PCR mutagenesis is displayed in Supplementary Table 3.

### Expression and purification of PC-PLCs
Plasmids, including various PC-PLC gene inserts, were transformed into the *Escherichia coli* BL21(DE3) strain. The transformed bacteria were cultured in the liquid Terrific Broth (TB) medium (24 g/l yeast extract, 20 g/l tryptone, 4 ml/l glycerol, 17 mM $KH_2PO_4$, 72 mM $K_2HPO_4$) at 37 °C with shaking at 180 rpm, until reaching the optical density $A_{600}$ of 0.8 to 1.0. Protein production was induced by addition of 0.4 mM isopropyl β-D-1-thiogalactopyranoside (IPTG) to the liquid medium and induced cells were further incubated at 20 °C and 180 rpm for 18 h. The cells were then centrifuged for 10 minutes at 4,000 *g*, the pellet resuspended in the lysis buffer (50 mM Tris-HCl pH 8.5, 500 mM NaCl, 10% v/v glycerol), and lysed by sonication. The cell debris was pelleted by centrifugation at 50,000 *g* for 60 minutes and the clarified lysate containing the cytoplasmic fraction was loaded onto 10 ml of chitin resin (NEB, USA) packed in a Tricorn 10/100 column (GE Healthcare, USA). All chromatographic steps were performed using the ÄKTA Purifier 100 (GE Healthcare, USA) at room temperature. The sample-loaded resin was washed with 10 column volumes of the wash buffer (20 mM Tris-HCl pH 8.5, 500 mM NaCl,), followed by 3 column volumes of elution buffer (20 mM Tris-HCl pH 8.5, 500 mM NaCl, 50 mM DTT). The flow was stopped overnight and the column was left at room temperature to induce thiol-mediated cleavage of the intein-chitin binding domain, which releases the protein of interest (various PC-PLCs) from the chitin matrix. After resuming the flow following the overnight incubation, the PC-PLC-rich fractions were eluted and collected. The collected fractions were pooled and concentrated in 15 ml 10,000 kDa cut-off Amicon Ultra centrifugal filters (Millipore, USA) and loaded onto a Tricorn 16/600 or 26/600 (*Lm*PC-PLC$^{C143S+C168K}$) Superdex 75 pg size exclusion column (GE Healthcare, USA), pre-equilibrated in the running buffer (20 mM Tris-HCl pH 8.5, 500 mM NaCl). The fractions containing proteins of interest, as evaluated by SDS-PAGE (NuPAGE 4-12% bis-Tris protein gel (Invitrogen, USA)), were

pooled, concentrated in 10,000 kDa cut-off Amicon Ultra centrifugal filters (Millipore, USA), aliquoted and stored at −80 °C. Re-run of SEC peak 3 (Fig. 1a, Supplementary Fig. 2d) was performed on the Tricorn 10/300 Superdex 75 GL. To determine the expected elution volume of PC-PLC, we performed a calibration of the HiLoad 16/600 Superdex 75 pg column using same buffer, and protein standards (IgG 150 kDa, transferrin 75 kDa, ovalbumin 43 kDa, scFvs 25 kDa, aprotinin 6.5 kDa).

### Expression and purification of LLO
Bacterial expression, cell lysis and supernatant collection were performed as for PC-PLCs, with an exception of the lysis buffer used (20 mM MES/NaOH pH 6.5, 150 mM NaCl, 10% v/v glycerol). The lysate with LLO was loaded onto 30 ml Ni-NTA Superflow resin (QUIAGEN, Germany) packed in a XK 16/200 column (GE Healthcare, USA), washed with wash buffer (20 mM MES/NaOH pH 6.5, 150 mM NaCl), and eluted with an imidazole gradient in 20 mM MES/NaOH pH 6.5, 150 mM NaCl, with 500 mM imidazole as the final concentration. Fractions containing LLO were pooled together and incubated with tobacco etch virus (TEV) protease (TEV:LLO molar ratio 1:50) during overnight dialysis in wash buffer, to remove the $His_6$-tag. After overnight incubation, the sample was loaded again on the nickel affinity column. The flow-through fraction containing LLO was collected, assessed for purity by SDS-PAGE, concentrated in 10,000 kDa cut-off Amicon Ultra centrifugal filters (Millipore, USA), aliquoted and stored at −80 °C.

### Electrophoresis
SDS-PAGE analysis was performed using NuPAGE Bis-Tris 4–12% gels (ThermoFisher, USA) according to the manufacturer's protocol. SDS-PAGE gels were stained with ProBlue Safe Stain (Giotto Biotech, Italy). Blue native-PAGE was performed using Novex Tris-Glycine 4–12% gels (ThermoFisher, USA), with 0.02% Coomassie Blue G-250 dye added to the cathode buffer.

### Edman sequencing
*Lm*PC-PLC wild-type samples were separated by SDS-PAGE. After electrophoresis, the gel was blotted onto polyvinylidene difluoride membrane (PVDF) using the iBlot2 system (ThermoFisher, USA). Blotting program ran for 7 minutes: 20 V for 1 minute, 23 V for 4 minutes and 25 V for 2 minutes. Following the transfer, the membrane was washed in 50% v/v methanol for 5 minutes, then stained in 0.1% w/v Coomassie Blue R-250, 50% v/v methanol, and 5% v/v acetic acid for 1 minute. The membrane was then de-stained with 50% v/v methanol and then air-dried. The protein bands were then excised from the membrane and sent for the N-terminal Edman sequencing. The sequencing was performed at Department of Molecular and Biomedical Sciences, Josef Stefan Institute (Ljubljana, Slovenia) on the Procise protein sequencing system 492 A (PE Applied Biosystems, USA). Edman chemistry using pulsed-liquid blot was performed for derivation of phenylthiohydantoin amino acids. Derivates were analyzed with high-performance liquid chromatography (HPLC) system 140 C (PE Applied Biosystems, USA) using RP C18 column Spheri-5 (5 μm, 220 mm by 2.1 mm) (Brownlee, USA).

### Circular dichroism
The circular dichroism (CD) spectra of proteins were measured with a Chirascan CD spectrometer (Applied Photophysics, UK) at wavelengths between 200 to 250 nm, with 1 mm of optic path in High Precision Cell quartz glass cuvettes (Hellma Analytics, Germany). The protein samples were diluted to 0.1-0.2 mg/ml with 10 mM $Na_2HPO_4$ pH 7.4. Five repeated measurements of the protein sample and blank (sample buffer diluted in 10 mM $Na_2HPO_4$ pH 7.4) were recorded and the averaged blank spectrum was subtracted from the averaged sample spectrum.

### Size-exclusion chromatography coupled with a multi-detection system consisting of a UV-detector and a multi-angle light scattering photometer (MALS) detector (SEC/UV-MALS)
Molar mass characteristics of the *Lm*PC-PLC were determined using SEC connected to a UV as concentration detector operating at a wavelength of 280 nm (Agilent Technologies, USA) and a DAWN multi-angle light scattering photometer (Wyatt Technology Corp., USA). Separations were performed at room temperature using a Tricorn 10/300 Superdex 75 GL SEC column (10 mm ID × 30.0 cm L, 8.6 μm particle size, 70 kDa exclusion limit, Sigma-Aldrich, USA). Buffer (500 mM NaCl and 20 mM Tris-HCl pH 8.5) with a flow rate of 0.5 ml/min was used as mobile phase. Toluene was used to calibrate the 90° LS detector, while the other detectors were normalized using Bovine Serum Albumin (BSA) as an isotropic scatterer (Pierce™ Bovine Serum Albumin Standard Ampules, ThermoFisher, USA). The injection volume was 100 μL. The specific refractive index increment (dn/dc) used to calculate the molar mass of *Lm*PC-PLC was 0.185 ml/g. Astra 8.0.2.5 software (Wyatt Technology Corp., USA) was used for data acquisition and analysis.

### Crystallization and structure determination of *Lm*PC-PLC[C143S+C168S]
1 μl of *Lm*PC-PLC[C143S+C168S] (12 mg/ml, in 20 mM Tris-HCl pH 8.5, 500 mM NaCl), was mixed with the 1 μl of a reservoir solution (18% v/v PEG 8000, 100 mM Na-citrate, 20% v/v propan-2-ol, 7% w/v trimethylamine-N-oxide, 5 mM phosphocholine), and crystallized using the hanging drop vapor diffusion method over 0.5 ml reservoir solution at 20 °C. Crystals were flash frozen and kept in liquid nitrogen until data acquisition. Diffraction data were collected at 100 K and the wavelength of 1.26 Å (close to absorption edge of Zn)[70] on the XRD2 beamline at Elettra Synchrotron (Trieste, Italy). The diffraction data was processed with the program XDSAPP[71] (version 2.0) up to a resolution of 2.0 Å (Supplemental Table 1). The crystal structure was solved by single-wavelength anomalous dispersion (SAD) phasing, utilizing the anomalous signal of the Zn ions present in the structure. The phasing was done with Autosol (version 1.19) and the automatic building of the first model with Autobuild using Phenix software package (version 1.20.1)[72]. The structure was further refined by iterative cycles of manual structure building in Coot (version 0.98)[73] and automated structure refinement (Phenix.refine, version 1.20.1)[72]. Validation and deposition was done on wwPDB server (www.wwpdb.org)[74]. The graphical presentation of structures in figures was done using Chimera X (version 1.6.1)[75] and Pymol2 (version 2.5.2)[76]. Additionally, data at 1.33 Å was collected in order to determine the chemical nature of ion species in the active site based on anomalous different maps of data collected at 1.26 Å and 1.33 Å X-ray wavelength.

### Inductively coupled plasma optical emission spectroscopy (ICP-OES) and inductively coupled plasma mass spectrometry (ICP-MS)
To determine the metal ion identity and quantify the molar ratio with the LmPC-PLC, we performed ICP-OES and ICP-MS analyses. 100 μL of the *Lm*PC-PLC sample (C143S + C168S mutant, 10 mg/ml) or blank (the background buffer of the protein sample after concentration of the sample using 15 ml 10,000 kDa cut-off Amicon Ultra centrifugal filters, Millipore, USA) were pipetted and weighed into a clean and dry ultrawave digestion vessel, followed by the addition of 4 ml $HNO_3$ and 1 ml $H_2O_2$. The vessel was closed according to the instructions for the ultrawave digestion oven (Ultrawave, Milestone, USA), and the digestion was carried out. Afterward, the solution was poured into a 10 ml volumetric flask, diluted to 10 ml volume with purified water (Elix 10 & Milli-Q Gradient Millipore, Bedford, USA), and mixed.

We first performed semi-quantitative analysis of the *Lm*PC-PLC sample and blank using ICP-OES (Varian 715-ES with autosampler Varian SPS-3, Agilent, USA). ICP-OES operating conditions were: power: 1.10 kW; plasma flow (Ar): 15.0 l/min; auxiliary flow (Ar): 1.50 l/min; nebulizer pressure: 200 kPa; pump rate: 15 rpm; viewing height: 12 mm; instrument stabilization delay: 15 s, and replicate read time: 5 s. Of the 50+ atomic species analyzed, only Fe, Na, Zn, Ca and S were detectable in the protein sample and of those Fe, Zn and S levels were significantly higher than in the blank.

Results of the semi-quantitative analyses thus indicated Fe and Zn as the ion candidates present in the active site of *Lm*PC-PLC. To determine Fe and Zn content we analyzed the samples using ICP-MS (Agilent 7500ce Series, with an octopole collision cell, Agilent, USA). 1 ml of the sample solution was pipetted into a 15 mL centrifuge tube, followed by the addition of 50 µl internal standard solution (Y, Sc, Ge, Gd [10 mg/l]), diluted to 10 mL volume with purified water, and mixed. Calibration solutions for elements were matrix-matched. The following isotopes were monitored: $^{57}$Fe and $^{66}$Zn. Operating plasma conditions were: RF power: 1550 W, tunable; sample depth: 10 mm, tunable; carrier gas: 0.90 l/min, tunable; makeup gas: 0.15 l/min, tunable; nebulizer pump: 0.1 rps, fixed; sample pump: 0.1 rps, fixed, and spray chamber temperature: 4 °C, fixed. Octopole collision cell conditions were: kinetic energy discrimination mode; energy discrimination: 16 V, and He collision gas flow: 5 ml/min. Acquisition parameters: points per peak: 1 and dwell time per mass: 0.3 s.

To determine the protein content, i.e., based on the sulfur content from four methionine residues in *Lm*PC-PLC amino acid sequence, we performed a quantitative ICP-OES measurement. The result was consequently used for the determination of the stoichiometry of metal ions bound to a *Lm*PC-PLC molecule. 5 ml of the sample solution was pipetted into a 15 mL centrifuge tube, followed by the addition of 50 µl internal standard solution (Y, Sc [100 mg/l]), and mixed. Calibration solutions for elements were matrix-matched. The following wavelengths were monitored: S 180.669 and S 181.972. ICP-OES operating conditions were the same as described above for semi-quantitative ICP-OES analysis. All quantitative measurements were conducted in triplicates.

### Liposome preparation
Multilamellar vesicles (MLVs) were made by preparing chloroform mixtures of lipids, purchased from Avanti Polar Lipids (USA): POPC (1-palmitoyl-2-oleoyl-glycero-3-phosphocholine), SM (brain sphingomyelin), and CHOL (cholesterol). Thin lipid films were produced by evaporation of chloroform on rotary evaporator for 2 h (Buchi, Switzerland). The dried lipid film was hydrated with buffer (20 mM MES-NaOH (pH 5.7 or 6.5) or 20 mM Tris-HCl (pH 7.4 or 8.5), 150 mM NaCl, 50 µM ZnSO$_4$) and subjected to 10 freeze-thaw cycles in liquid nitrogen to form 30 mM MLVs. MLVs were stored at −80 °C for up to one month. Large unilamellar vesicles (LUVs) were produced by the extrusion of MLVs through a lipid extruder LiposoFast (Avestin, Canada) with a 200 nm pore size filter. LUVs were stored at 4 °C before use. LUVs older than 7 days were discarded. The lipid concentration and composition of MLVs and LUVs were estimated by a colorimetric assay (WAKO diagnostics, Japan).

### Malachite green phosphate assay
The enzymatic activity of PC-PLCs was evaluated by a colorimetric test. In this test, we used the alkaline phosphatase to release the free phosphate from phosphocholine molecules, produced by PC-PLC lipid hydrolysis of MLVs. 100 µl reaction mixtures were prepared in 1.5 ml tubes by diluting PC-PLC to a final concentration of 50 nM and MLVs to 4.5 mM with reaction buffer 20 mM MES-NaOH for pH 5.7 or 6.5, and 20 mM Tris-HCl for pH 7.4 or 8.5, 150 mM NaCl, with 50 µM ZnSO$_4$, as indicated in figures. Reaction mixtures were briefly mixed by pipetting, spun down, and incubated at 37 °C for 1 h. After incubation, the tubes

with reaction mixtures were centrifuged at 16,000 *g* for 10 minutes, to pellet the MLV substrate. 10 µl of the solution containing phosphocholine was transferred to a 96-well plate (Corning, USA) and 70 µl alkaline phosphatase solution (Sigma-Aldrich, USA) with a final concentration of 40 U/ml was added to each well. Negative controls were prepared using the same procedure but without PC-PLC addition. The plate was then incubated at 37 °C for 30 minutes. After incubation, 20 µl of reagent (Malachite Green Phosphate Assay Kit, Sigma-Aldrich, USA) was added. The plate was incubated for 30 min at room temperature and the absorbance at 630 nm was measured with a microplate reader (BioTek, USA) using Gen5 software (version 3.11). Three independent measurements were performed (*n* = 3), using distinct samples.

### O-(4-Nitrophenylphosphoryl) choline (4-NPPC) enzymatic assay
4-NPPC (Sigma-Aldrich, USA) was dissolved in ultrapure water to the stock concentration of 20 mM. The assay was performed in 96-well plates (Corning, USA). In each well, 4-NPPC was diluted with buffer (20 mM MES-NaOH (pH 5.7 or 6.5), or 20 mM Tris-HCl (pH 7.4 or 8.5), 150 mM NaCl, 0 or 50 µM ZnSO$_4$) to 1 mM concentration and a final volume of 198 µl. 2 µl of PC-PLC solution was added to a final concentration of 500 nM (or 5 µM, Supplementary Fig. 13c. Negative controls were done using the same procedure but without PC-PLC addition. The absorbance was measured with a microplate reader (BioTek, USA) using Gen5 software (version 3.11) at 405 nm (highest signal/noise ratio) or at 347 nm when comparing activities in buffers of different pH (isosbestic point). Absorbance measurements were performed at 37 °C, every 5 minutes for 4 hours. Using linear regression, slopes representing the rate of reaction were obtained. Three independent measurements were performed (*n* = 3), using distinct samples.

### Lipid sedimentation assay
Reactions were prepared by diluting MLVs into the buffer (20 mM MES-NaOH (pH 5.7 or 6.5), or 20 mM Tris-HCl (pH 7.4 or 8.5), 150 mM NaCl, with 50 µM ZnSO$_4$, as indicated) to a final MLV concentration of 8 mM. Then PC-PLC was added to the final concentration of 1.4 µM and reactions were mixed and incubated for 30 minutes at room temperature. The negative controls were prepared in the same way, without the addition of PC-PLC (LLO only). After 30 minutes of incubation, LLO was added to the final concentration of 3 µM, mixed by pipetting and incubated for 30 minutes at room temperature. The mixture was centrifuged (10 min, 16,000 *g*) and the supernatant was transferred to fresh tube and stored for analysis on SDS-PAGE gel. The pellet was washed once with reaction buffer followed by centrifugation (10 min, 16,000 *g*). Pellet and supernatant were loaded onto SDS-PAGE gel, followed by staining with ProBlue Safe Stain. Three independent measurements were performed (*n* = 3), using distinct samples. The intensity of the bands and the ratio of bound to unbound protein was quantified with GelQuant.NET (version 1.8.2) biochemlabsolutions.com, USA).

### Hemolysis
Bovine erythrocytes, stored at 4 °C in Alsever's preservative (2.05% dextrose, 0.8% Na-citrate, 0.055% citric acid, and 0.42% NaCl), were centrifuged at 800 *g* at room temperature. The supernatant was discarded and fresh reaction buffer (20 mM MES-NaOH (pH 5.7 or 6.5, 150 mM NaCl) or 20 mM Tris-HCl (pH 7.4 or 8.5), 150 mM NaCl, 50 µM ZnSO$_4$) was added and exchanged 3-4 times. The suspension was diluted to the optical density of 1 measured at 630 nm. 90 µl of erythrocyte suspension was added to the wells of a clear 96-well plate (Corning, USA) and 10 µl of *Lm*PC-PLC (different concentrations) were added. Then 100 µl of LLO (0.5 nM) was added with automatic multichannel pipette (Biohit, Finland) and the plate was read immediately after addition with Synergy™ Mx plate reader (BioTek, USA) using Gen5 software (version 3.11). Absorbance measurements at 630 nm

were made every 30 seconds for 20 minutes, with constant shaking in between measurements. Three independent measurements were performed ($n = 3$), using distinct samples.

## Fluorophore release from lipid vesicles

5(6)-Carboxyfluorescein (CF, Sigma-Aldrich, USA) loaded LUVs were prepared using buffer (50 mM CF, 20 mM MES-NaOH, pH 6.5, 150 mM NaCl, 50 µM ZnSO$_4$). The dye surrounding LUVs was removed on a 10 ml gravity flow Sephadex G-50 column (GE HealthCare, USA). Fractions containing vesicles loaded with dye were pooled together, their phospholipid and cholesterol content were estimated by colorimetric enzymatic assay (WAKO diagnostics, Japan) and stored at 4 °C. The test was performed in 96 well black opaque bottom plates and fluorescence reading (495 nm excitation, 517 nm emission) was performed on Synergy™ Mx plate reader (BioTek, USA) using Gen5 software (version 3.11). 250 µM of LUVs of indicated lipid composition were resuspended to 200 µl final volume in each well and placed in the reader to measure the kinetics of LUV alone (baseline). After 30 minutes, 1 µl PC-PLC was added up to a final concentration of 0.5 or 2 µM. After a further 30 minutes of kinetic reads 1 µl of LLO was added, 0.5 µM final concentration. The release of CF due to LLO was measured for 90 minutes and then Triton X-100 detergent (2 mM final concentration) was added to achieve 100% CF release. Three independent measurements were performed ($n = 3$), using distinct samples. The data was expressed as a percentage of permeabilized vesicles before addition of Triton X-100.

## Protein structure prediction

For protein structure prediction, we used ColabFold[77], a protein structure and complex prediction tool, which uses AlphaFold2 and Alphafold2-multimer, and is available through the Google Collaboratory. For the structure prediction, no template information was used as an input, only the amino acid sequence of pro*Lm*PC-PLC without the signal peptide (Uniprot ID: P33378). Multiple sequence alignment (MSA) mode was MMseqs2 (Uniref+Environment) and pair mode was paired+unpaired, AlphaFold2-ptm was used to run the structure prediction, number of recycles was set to 3, and 5 structure prediction models were generated.

## Concentration dependent inhibition of *Lm*PC-PLC by peptides

Lyophilized peptides NE-6 (NACCDE) and NS-26 (NACCDEYLQT-PAAPHDIDSKLPHKLS) were purchased from Proteogenix (France). Obtained lyophilized peptides were resuspended in 500 µL of ultra-pure water to 8 mg/ml peptide solution, aliquoted, and frozen until use at −80 °C. Enzyme inhibition by the peptides was measured using the 4-NPPC enzymatic assay. We tested the activity of *Lm*PC-PLC at different peptide concentration (0-1000 µM for NE-6 and 0-250 µM for NS-26). Relative activity was calculated as ratio of the activity with the peptide present compared to the activity of PC-PLC without peptide added. Three independent measurements were performed ($n = 3$), using distinct samples. Four-parameter logistic curve was fitted in GraphPad Prism software (version 8.4.2, USA) and IC50 parameter and 95% CI was determined from the fit.

## Peptide inhibition assay on homologs and cysteine mutants

Inhibition of the *Lm*PC-PLC homologs and cysteine mutants by peptides NE-6 and NS-26 was measured using the 4-NPPC enzymatic assay or malachite green phosphate assay (using 100% POPC MLVs), as described above, with the addition of different concentrations of the peptides. For the malachite green phosphate assay the reaction consisted of 10 µM peptides, 50 nM PC-PLC, and 50 µM ZnSO$_4$. For the 4-NPPC assay 100 µM of peptides, 500 nM of PC-PLC, and 500 µM ZnSO$_4$ was used in reaction. Three independent measurements were performed ($n = 3$), using distinct samples. Relative activity corresponds to the activity with the peptide present compared to the activity of PC-PLCs without peptide added.

## Transmission electron microscopy at cryogenic conditions (cryo-EM)

LUVs of different lipid compositions were prepared from MLVs as described above. Experiments with LLO only were performed at different incubation times (30 min, 2 h and 6 h), at 20 °C or 37 °C and pH 6.5. For experiments with LLO and *Lm*PC-PLC preincubation, all the incubation steps were performed at 37 °C. 2.5 mM LUVs were preincubated with 5 µM *Lm*PC-PLC for 30 minutes. After that, LLO was added to a final concentration of 5 µM and the sample mixtures were further incubated for 60 minutes. After incubation, 3 µl of each sample was transferred to glow-discharged (GloQube® Plus, Quorum, UK) Quantifoil R1.2-1.3 grids (Quantifoil, Germany) and blotted under 100% humidity using Mark IV Vitrobot (Thermo Fisher Scientific). Blot force used was 3 and blot time was 6.5 s. Micrographs were collected on cryo-transmission electron microscope Glacios (Thermo Fisher Scientific) operated at 200 kV and equipped with Falcon 3 direct electron detector (Thermo Fisher Scientific), at a nominal magnification of 5,300x and 73,000x and defocus range of −3 µm to −4 µm.

## Statistical analysis

*P* values were determined in GraphPad Prism software (version 8.4.2, USA). Two-sided Dunnett's multiple comparisons test[78] following 2-way ANOVA was performed when comparing multiple experiments or conditions to the control (Fig. 1c, f, g, Fig. 2a, Fig. 3c, d, Fig. 4b, c e-h, Fig. 5c, d, e, h, Supplementary Fig. 14b, Supplementary Fig. 16a). No additional adjustments were made for multiple comparisons. Two-sided Student's t-test was performed when comparing one experiment to the control (Fig. 3b, Fig. 5a, b, Supplementary Fig. 12f, Supplementary Fig. 13c, Supplementary Fig. 17b). Normal distribution was always assumed. Determination of EC$_{50}$ values (Fig. 4d) was done using non-linear fit (fitting a 4-parameter sigmoidal curve) using top constrain ($=1$).

## Reporting summary

Further information on research design is available in the Nature Portfolio Reporting Summary linked to this article.

## Data availability

The crystal structure of *Lm*PC-PLC was deposited to wwPDB[74] database under the code PDB-ID 8CQM. Raw data crystallographic data is linked to the PDB entry and is stored on XRDa database (https://doi.org/10.51093/xrd-00127). Additional data is provided in the Supplementary information file. Source data are provided with this paper.

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

## Acknowledgements
We thank the Slovenian Research Agency for funding (grant numbers J1-9174 (M.P., A.K.), P1-0391 (G.A., M.P., M.A., A.K.), P2-0145 (E.Ž.), P1-0034 (S.B.H.) and IO-0003 (M.P.) and the PhD fellowship for N.P.). We thank the colleagues at the Veterinary Faculty, University of Ljubljana for isolates of genomic DNA of *Listeria monocytogenes*. Authors would like to thank Amela Kujović for the help with the preparation of W1 mutants and Iris Štucin and Breda Novak, and Mirjana Širca for the technical support with ICP-OES/ICP-MS and SEC-MALS analyses, respectively. We also thank the Centre of Excellence for Integrated Approaches in Chemistry and Biology of Proteins (CIPKeBiP), Ljubljana, for the initial crystallization screening as well as the staff at the Elettra synchrotron, Trieste, Italy, for the valuable technical help with X-ray diffraction data collection.

## Author contributions
M.P. conceived the project. N.P., M.P., M.A. and A.K. conceived and planned the experiments. N.P., M.A. and A.K. carried out the experiments. N.P., M.P., M.A., G.A., S.B.H., E.Ž. and A.K. contributed to the interpretation of the results. M.P. and N.P. wrote the manuscript and M.A., A.K. and G.A. reviewed the manuscript and contributed to the finalization of the text and figures. All authors provided critical feedback and helped shape the research, analysis and manuscript.

## Competing interests
The authors declare no competing interests.

## Additional information

**Peer review information** : *Nature Communications* thanks Yu-Huan Tsai, and the other, anonymous, reviewers for their contribution to the peer review of this work. A peer review file is available.

