## [Peer Review File · Nature Communications]

REVIEWER COMMENTS

Reviewer #1 (Remarks to the Author):

In this study the authors provide crystal structures of LmPC-PLC, the phospholipase critical for *Listeria monocytogenes* infection. They further performed sophisticated biochemical experiments to demonstrate that multiple mechanisms are adopted by *L. monocytogenes* in regulating LmPC-PLC enzymatic activity, including Zn²⁺ dependence and oligomerization. Furthermore, the crystal structures provide insights in understanding the potential implication of flexible loops in modulating Zn²⁺ binding (Zn1), and therefore the enzymatic activity. The AlphaFold2 analysis combined with biochemical analysis further indicated the involvement of N-terminal propeptide in inhibiting Zn1 and thus enzymatic activity of LmPC-PLC. While the authors clearly demonstrated how LmPC-PLC acts differently as compared to other PLCs via structural and biochemical characterization, how these regulatory mechanisms lead to host cell membrane damage by cooperating with LLO and achieve successful infection is relatively weak. This can affect the application of LmPC-PLC regulatory mechanisms in clinical treatment. Nevertheless, the findings in the study have provides substantial insights in understanding the molecular mechanism of LmPC-PLC activation. Other specific concerns are listed below.

L81. Achieved “by” releasing the

L220 (Supplementary Fig. 7c). Would it be possible to use TPEN, which is more specific to Zinc chelation? Better to have the additional experimental setting Zn + Zn chelator as a control.

L304 (Fig. 4i). The inhibitory function of NE-6 seems to depend on assay substrates, where NE-6 clearly inhibited enzymatic activity against 4-NPPC but not POPC. Whereas, NS-26 works well in both systems. Any explanation on this?

L306. Would it be possible to perform the enzymatic activity experiments with NS-26 cysteine replacement mutant peptide to confirm the implication of disulfide bond formation between C143 and the propeptide in inhibitory function of the propeptide?

L337 (Fig. 5G). C143 was demonstrated to inhibit enzymatic activity by interacting with N-ter propeptide (Fig. 4), but C143S showed no difference in modulating LLO-mediated vesicle permeabilization. Any explanation on this?

L342-343 (Fig. 5e). Not very evident with very small difference (21% to 19%). Is it Student T test for the comparison?

L346-348 (Supplementary Fig. 13b). Does not seem to be the case for Bc PC-PLCs. SM appears to reduce BcPC-PLC-mediated LLO binding to the lipid. Please explain.

L400-401. It would be great to observe the high content of LLO at the sea urchin-like cluster by immunogold staining.

L407-409. Does LmPC-PLC change the action mode of LLO or increase LLO binding to the lipid due to higher exposure of cholesterol?

Could higher conc. of LLO alone achieve the arc formation?

Any evidence demonstrating the direct interaction between LLO and LmPC-PLC?

Is LmPC-PLC-mediated lipid processing along sufficient to increase LLO binding?

L461-464. Does it mean that the self-inhibiting activity of LmPC-PLC may contribute to higher infection efficiency? It is thus important to test the role of propeptide-mediated LmPC-PLC inhibition at least in cell-to-cell spread in vitro experiments before proposing its application for treatment.

Reviewer #2 (Remarks to the Author):

Petrisic and co-workers report the first structure (and a functional study) of a phosphatidylcholine-specific phospholipase C (PC-PLC) from *Listeria monocytogenes* (Lm), the causative agent of the foodborne disease listeriosis. The authors show compelling evidence to support a model of the synergy between LmPC-PLC activity and the pore-forming activity of pore-forming toxin listeriolysin O (LLO). Several regulation mechanisms are suggested including through oligomerization and inhibition by its pro-peptide. Altogether, this work is providing novel insights into the structure and function of LmPC-PLC and its role in infection by *Listeria monocytogenes*. The data look compelling and the conclusions justified. I only have a few comments.

Plasticity of the active site (p.11-12): I do not understand how the authors reconcile the unusual position of D55 with its crucial role in catalysis. I recommend that the authors clarify that point.

The authors discuss the difference in the conformation of the loops D55-S64 and R75-F83 in MolA and MolB, and also in comparison to Bc and CpPC-PLC. In particular the discussion of the role of these loops in occluding the active site is very interesting. I am surprised though that the authors do not consider, in this discussion, the potential role of these loops in the interaction of LmPC-PLC with membranes. Is it conceivable that these loops would be stabilized in their open state through interactions with membrane lipids? Or is it known that these loops are not involved in membrane binding. If so, which ones are?

The surface of BcPC-PLC is rich in tyrosines and tryptophanes and it has been shown, for BtPI-PLC (Grauffel et al, JACS, 2014) and spider venom GDPD-like phospholipases D (SicTox, Moutoussamy et al, PLoS Comp Biol, 2022), that tyrosines and tryptophanes engage in specific interactions with PC lipids (including the formation of aromatic cages around choline groups). The alignment of the Lm and Bc PC-PLC sequences show that several of Bc Tyr and Trp are conserved in the Lm enzyme, including in the loops 55-64 and 75-83. Are these solvent-exposed in the Lm PC-PLC?

The pLDDT scores (per residue) should be reported for the AlphaFold model of the proenzyme, so that the reader can evaluate the quality of the model in the propeptide region. Values below 70 should be treated with caution.

Reviewer #3 (Remarks to the Author):

Given my expertise, I have been primarily invited to comment on the crystallographic content of this manuscript. However, since I also have extensive expertise in bioinorganic and protein chemistry, I have made additional comments related to those aspects of the manuscript.

Crystallography: I only have access to the validation report (not coordinates or maps). The data listed in Table 1 (SI) indicate a resolution cut-off for the data of 2.0 angstroms, however the $I/\sigma I$ in the highest bin = 4.19 (2.072 - 2.0 angstroms), which indicates the data have been artificially truncated and may have been processed to higher resolution. Can the authors revisit the data processing to see if this might be possible?

The data were collected using X-radiation of wavelength 1.26 angstroms, which is at the Zn K-edge. This is sensible for structure solution, but these data will probably suffer from some radiation damage due to absorption by the Zn atoms in the crystal. A more appropriate method would be to collect a 'native' dataset away from the edge (eg. at 13.0 keV) and use those data for refinement. I would also like to see anomalous difference Fourier electron density maps calculated from data collected at an energy 'below' the Zn-edge. These maps should show a lack of anomalous signal and then confirm the metal atoms unequivocally as Zn. At the moment, the metal atoms in the structure could be Zn, Co, Ni, Cu or Fe, based on the fact that anomalous signal was detected from the data collected at 1.26 angstroms.

I find the description of the Zn coordination lacking in this manuscript, both in text and in Figure 3a (inset). The 'reorientation' of residue D55 seems to correlate with decreased occupancy of the Zn1 site, however the other putative ligands seem to be in place. There is no discussion on what interactions might be contributing to the altered orientation of D55, therefore, it is very difficult to understand the differences between this structure and homologues. D55 is obviously crucial for activity, which makes this more perplexing. The activity of the enzyme is measured in the presence of various divalent cations (Fig 2a), but there doesn't seem to be an attempt to determine the metal:protein stoichiometry (eg. by ICP-MS) and/or affinity (ITC). Is the Zn1 site occupancy lower than Zn2 and Zn3 in the protein in solution or only in the crystal?

The data presented in Figure 1 and S2 regarding the estimates molecular weight of the LmPC-PLC protein are confusing at best. Apparently the dotted line in Fig 1a represents an elution volume of 66 mL, which corresponds to a 28 kDa globular protein, yet peaks 1 (close to the 28 kDa mark) and 2 (eluting after the 28 kDa mark) are described as oligomers. Peak 3 elutes well after this mark and therefore appears to have a molecular weight of around 10 kDa, however this is not estimated. This section needs to be revisited and revised. A preparative column (Fig 1) should never be used for the estimation of molecular weight - at least an analytical column should be employed and more preferably a technique such as MALLS or AUC. At the moment, the analysis is not well described. It is also used to support the crystal structure as consisting of 'monomers' even though the crystal packing shows associations between the protein units in the crystal.

Overall, from the point of view of structural biology, bioinorganic and protein chemistry, the above points need to be addressed in full before this manuscript is suitable for publication.

**Point-by-point response to the comments of the reviewers for the revised manuscript NCOMMS-23-11000 by Petrišič et al.**

For clarity, the reviewers' comments are in **blue** and in *italics*. Authors' responses are in regular black text.

Response to Reviewer #1:

In this study the authors provide crystal structures of LmPC-PLC, the phospholipase critical for Listeria monocytogenes infection. They further performed sophisticated biochemical experiments to demonstrate that multiple mechanisms are adopted by L. monocytogenes in regulating LmPC-PLC enzymatic activity, including Zn²⁺ dependence and oligomerization. Furthermore, the crystal structures provide insights in understanding the potential implication of flexible loops in modulating Zn²⁺ binding (Zn1), and therefore the enzymatic activity. The AlphaFold2 analysis combined with biochemical analysis further indicated the involvement of N-terminal propeptide in inhibiting Zn1 and thus enzymatic activity of LmPC-PLC. While the authors clearly demonstrated how LmPC-PLC acts differently as compared to other PLCs via structural and biochemical characterization, how these regulatory mechanisms lead to host cell membrane damage by cooperating with LLO and achieve successful infection is relatively weak. This can affect the application of LmPC-PLC regulatory mechanisms in clinical treatment. Nevertheless, the findings in the study have provides substantial insights in understanding the molecular mechanism of LmPC-PLC activation. Other specific concerns are listed below.

L81. Achieved “by” releasing the

Thank you for noticing the typo. It has been corrected in the revised manuscript.

L220 (Supplementary Fig. 7c). Would it be possible to use TPEN, which is more specific to Zinc chelation? Better to have the additional experimental setting Zn + Zn chelator as a control.

We thank the reviewer for pointing out another potent chelator of Zn²⁺. We have done the experiments with TPEN as suggested. We have performed experiments with serial dilutions of TPEN, which was then added to LmPC-PLC supplemented with Zn²⁺ (50 μM). To compare the two metal chelators (TPEN and O-phenanthroline (O-phe, which was originally shown in the manuscript)) we also performed a similar experiment with O-phe. The results are displayed on the Supplementary Fig. 10, added text in the revised manuscript, lines 255-257. These results revealed that TPEN indeed appears to be a stronger inhibitor than O-phe. While TPEN has been used in Zn²⁺ treatment studies, it has been known to also chelate other metals and that Zn²⁺ chelating activity is also depended on the concentration of Zn ions (Cho et al., 10.4162/nrp.2007.1.1.29). Nevertheless, using TPEN in our assays complements other experiments (ICP-MS, ICP-OES and X-ray crystallography, please see the revised manuscript and answers to the Reviewer 3), where we demonstrate that LmPC-PLC does contain Zn ion in the active site (in addition, we now also show the presence of Fe ions in the active site), but the activity is dependent only on the presence of Zn ions.

To address the second part '*Better to have the additional experimental setting Zn + Zn chelator as a control*', we repeated the experiments previously shown on Supplementary Fig 7c, and performed O-phe inhibition assay also with LmPC-PLC supplemented with Zn ions. 50 μM Zn²⁺ was added to the protein and after 30 min of preincubation the enzymatic activity assay was performed with or without O-phe (1 mM). Also in this case, O-phe completely inhibited LmPC-PLC activity. The updated results are in Supplementary Fig. 9c.

L304 (Fig. 4i). The inhibitory function of NE-6 seems to depend on assay substrates, where NE-6 clearly inhibited enzymatic activity against 4-NPPC but not POPC. Whereas, NS-26 works well in both systems. Any explanation on this?

NE-6 is generally a weaker inhibitor than NS-26, for both substrates (Fig. 4d-h). This suggests that the longer peptide NS-26 is required for more efficient inhibition of *LmPC-PLC*, likely due to more efficient occlusion of the active site cleft in the case of the longer peptide NS-26. This was included in the main text lines 338-340.

L306. Would it be possible to perform the enzymatic activity experiments with NS-26 cysteine replacement mutant peptide to confirm the implication of disulphide bond formation between C143 and the propeptide in inhibitory function of the propeptide?

We thank the reviewer for the suggestion. While this would be possible (several peptides synthesized, single mutations of each of the two Cys in the propeptide and the simultaneous mutation of both Cys residues), we believe that we have already thoroughly addressed the importance of the amino acid sequence of both, the propeptide and the mature protein in the manuscript. The possibility that the propeptide forms the disulfide bond with the mature part of the enzyme was tested by using two *LmPC-PLC* propeptide based peptides, NE-6 and NS-26 on various forms of the mature *LmPC-PLC*: wild type *LmPC-PLC*, and Cys mutants (two Cys residues on the mature part of the enzyme were mutated): *LmPC-PLC*^{C143S}, *LmPC-PLC*^{C168S} and *LmPC-PLC*^{C143S+C168S} (now Fig. 4g, h). This indicated, that C143 on the surface of the mature *LmPC-PLC*, in the vicinity of the active site might play a role in inhibition by the peptides, possibly through formation of the disulfide bond. Mutating the proteins enabled us to check that mutated proteins retained the structure as well as the activity upon mutations, making sure that the effect comes purely from the change of the side chain. Importantly, there seems to be other factors, such as the steric occlusion of the active site cleft by the peptide that contribute to the inhibition besides the disulfide bond formation between the propeptide and the mature part of the enzyme (Fig. 4 g, h), as there is still some residual inhibition present in C143S. We also showed that the propeptide is specific to *LmPC-PLC*, as no significant inhibition has been shown for *Bc-* or *CpPC-PLC* (Fig. 4 e, f), additionally highlighting that the amino acid sequences of both, the propeptide as well the mature enzyme, are important.

L337 (Fig. 5G). C143 was demonstrated to inhibit enzymatic activity by interacting with N-ter propeptide (Fig. 4), but C143S showed no difference in modulating LLO-mediated vesicle permeabilization. Any explanation on this?

Two different sets of experiments were performed to pursue distinct aspects of the *LmPC-PLC* functioning ((i) inhibition of the mature *LmPC-PLC* by the (pro)peptide added *in trans*), Fig. 4, and (ii) impact of the phospholipase activity of *LmPC-PLC* on LLO pore formation (Fig. 5)). In the experiments described in Fig. 5 neither did we work with the pro-form of *LmPC-PLC* neither were the peptides NE-6 or NS-26 involved in the assay with the mature *LmPC-PLC*. Only the mature form of *LmPC-PLC* (wild type or mutants) was used. *LmPC-PC* Cys mutant C143S has enzymatic activity comparable to the wild type enzyme (Fig. 1f, g), so it makes sense that it makes no difference in LLO-mediated vesicle permeabilization (Fig. 5g). To further improve clarity, we added the label 'mature' in the title of Fig. 5.

L342-343 (Fig. 5e). Not very evident with very small difference (21% to 19%). Is it Student T test for the comparison?

We thank the reviewer for this comment. We agree that although statistically significant, this difference is probably not biologically significant as the difference is small. We have edited this part of the manuscript accordingly, i.e. deleted this part of the text: 'SM interacts with the membrane CHOL, forming distinct membrane lipid domains, which has been shown to decrease the availability of CHOL for binding to proteins⁵⁰. This was evident from the lower binding of LLO to the three-component membrane compared to binding to POPC:CHOL, molar ratio 80:20 (Fig. 5e).

L346-348 (Supplementary Fig. 13b). Does not seem to be the case for BcPC-PLCs. SM appears to reduce BcPC-PLC-mediated LLO binding to the lipid. Please explain.

Thank you for the comment. Previous Supplementary Fig. 13b, now Supplementary Fig. 17b, shows enzymatic activity of the PC-PLC homologs at two conditions (two different lipid membranes of MLVs) and not the effect on LLO binding. Although the activity of BcPC-PLC is impacted more by addition of SM, compared to Lm- or CpPC-PLC, at the conditions used in our experiment, BcPC-PLC still exhibited enough activity to enhance LLO binding to a similar extent as LmPC-PLC (and Cp homolog) as shown in Fig. 5e.

L400-401. It would be great to observe the high content of LLO at the sea urchin-like cluster by immunogold staining.

Thank you for this comment. The accumulation of LLO on the membrane preincubated with LmPC-PLC is high and the LLO clusters are compact, which may even prevent successful binding of antibodies. Moreover, at least in our experience, it is rather challenging to obtain antibodies that would work efficiently on LLO. We do agree and hope, however, that the results of our study will encourage future in depth studies of changes in dynamics, structure and (potentially) the mechanism of LLO binding and pore formation upon addition of the phospholipase or change of environmental conditions (lipid content, time, temperature, pH).

L407-409. Does LmPC-PLC change the action mode of LLO or increase LLO binding to the lipid due to higher exposure of cholesterol?

Thank you for this comment. Based on our results we believe that LmPC-PLC enhances LLO binding due to the increased exposure of cholesterol, which is the receptor of LLO, as a result of LmPC-PLC enzymatic activity. At conditions used in our experiments, we see that LLO binds to membranes pre-incubated with LmPC-PLC significantly more than in the absence of the phospholipase, most probably due to freshly exposed cholesterol similarly as it was described for CpPC-PLC and PFO. (Moe & Heuck, 10.1021/bi1013886, Flanagan et al., 10.1021/bi9002309). Thus, the mechanism of pore formation by LLO may not change, but rather the amount of the bound LLO and the proportion of different LLO populations (membrane bound monomers, membrane bound oligomers, membrane inserted LLO oligomers). We agree that the original title of this section 'LmPC-PLC influences the mode of membrane disruption by LLO' may not be appropriate, therefore we changed to 'LmPC-PLC influences the morphology of LLO oligomers on lipid vesicles', line 433.

Could higher conc. of LLO alone achieve the arc formation?

The arc is the common form of LLO oligomers that we and others have observed when LLO alone was added to lipid membranes (i.e., in the absence of LmPC-PLC). Arcs form on the membrane upon oligomerization of LLO and can also insert into the membrane thereby forming a functional pore, which is lined on one side by protein and on the other side by the lipid, as shown using AFM (Podobnik et al (10.1038/srep09623) and Ruan et al. (10.1371/journal.ppat.1005597)). These arcs merge with time and form pores, sometime ring-shaped or even bigger (merged pores with arcs), as has been shown before (Podobnik et al., cited above). In our current manuscript, at the conditions of the experiment shown in Fig. 6, we can also observe mainly pores using cryoEM, when membranes are exposed only to LLO. To show the presence of LLO arcs in the absence of LmPC-PLC, we added a Supplementary Fig. 18, where we show time dependent formation of oligomers and pores by LLO only, that include different incubation times and temperatures. However, when we preincubated membranes with LmPC-PLC and then added LLO (Fig. 6), the round shaped LLO pore could be hardly observed, while we still do observe arcs. We suspect that because of the overabundance of LLO on the membrane (due to the increased binding caused by LmPC-PLC exposed free cholesterol), there is not enough space for the arcs to grow or merge into round shaped pores, hence arcs as well as high load of bound proteins still

effectively disturb the vesicle membranes. We included this into the text of the revised manuscript, lines 467-470.

Any evidence demonstrating the direct interaction between LLO and LmPC-PLC?

To our knowledge, direct interaction of LmPC-PLC and LLO has not been shown yet, and our experiments also do not indicate any direct interaction. While we cannot exclude direct interaction between LLO and LmPC-PLC, it seems (as we have shown here) more plausible that the phospholipase activity of LmPC-PLC that facilitates exposure of the LLO receptor cholesterol is behind the synergy between LmPC-PLC and LLO. It has been shown in our manuscript that the phospholipase activity is crucial for the increased LLO binding by using various LmPC-PLC mutants, the inhibitors and PC-PLC activity and PC-PLC homologs (*Bc* and *Cp*) (Fig. 5c-g).

Is LmPC-PLC-mediated lipid processing along sufficient to increase LLO binding?

Based on our data presented in this manuscript, the phospholipase activity of LmPC-PLC is absolutely required and thus sufficient for the enhanced LLO binding, similarly as was shown with a homologous pair from *C. perfringens*, CpPC-PLC and PFO (Moe & Heuck, 2010, 10.1021/bi1013886).

L461-464. Does it mean that the self-inhibiting activity of LmPC-PLC may contribute to higher infection efficiency? It is thus important to test the role of propeptide-mediated LmPC-PLC inhibition at least in cell-to-cell spread in vitro experiments before proposing its application for treatment.

This comment addresses two separate matters. One part of our results suggested that the unique molecular features of the 'mature' (and not the pro-enzyme) LmPC-PLC might affect the *Lm* virulence by maintaining a balance between sufficient membrane dissolution of the phagosome and destruction of *Lm* cellular niche. Structural plasticity of the active site (flexibility of the active site loops), highly Zn²⁺-dependent activity and tendency to form inactive oligomers could be all implicated in this mediation of LmPC-PLC activity. Also, unlike *Bc*PC-PLC and *Cp*PC-PLC, which both have pI values below pH 6 (5.99 and 5.16 respectively), the pI value of LmPC-PLC is at pH 7.8. This could make it even more prone to oligomerization or aggregation in the cytosol. Purification of the recombinant LmPC-PLC below pH 8.5 or at lower salt concentration also resulted in lower resolution on SEC and lower yields, demonstrating relative instability at physiological conditions (Supplementary Fig. 2a). Similarly, LLO is also largely unstable in the cytosol due to its structural characteristics. At neutral pH of the cytosol it aggregates, annulling its membrane disrupting action inside the cytosol (Schuerch et al., 2005, 10.1073/pnas.0500558102, Bavdek et al., 2012, 10.1111/j.1742-4658.2011.08405.x). We added this comment to Discussion, lines 499-501.

The role of the propeptide is a different kind of regulation of LmPC-PLC *in vivo*. LmPC-PLC is expressed as a pre-pro-enzyme. Marquis and Hager (2000, 10.1046/j.1365-2958.2000.01708.x) showed that only upon cleaving of the propeptide can LmPC-PLC be secreted from the periplasmic space through the bacterial cell wall to the outside (phagosome lumen). This secretion is additionally regulated by proteolytic cleavage by the metalloprotease from *Lm* (Mpl) in an acidic pH and this enables a release of large quantities of the accumulated LmPC-PLC in a short time. However, this process is spatially and temporally separated from the processes in the host phagosome and later cytosol, discussed above. The fact that the protein is synthesized in the form of the inactive pro-enzyme gave us an idea that once active in the host cell, the mature LmPC-PLC could be inhibited by the addition of peptides (imitating) its propeptide. Therefore, we designed peptides that would have the sequence of the propeptide (two different length) and test whether they act as (specific) inhibitors added *in trans*, our results do indeed show this (Fig 4). Using peptides based on the propeptide directly (without any modifications to increase stability etc.) in cell-to-cell spread assays would be challenging as they are hard to introduce in the cell and are prone to digestion by host peptidases. We thank the reviewer for this comment, and agree that more separate studies (out of the scope of this manuscript) should be done to claim the applications.

OP) NATIONAL INSTITUTE OF CHEMISTRY

SI-1001 Ljubljana
Hajdrihova 19, POBox 660
Phone: +386 (0)1/476 02 00
Fax: +386 (0)1/476 03 00
<http://www.ki.si>

Therefore, we removed these propositions that that peptides could be used as therapeutics in *Lm*-related diseases from the text.

**Response to Reviewer #2:**

Petrišič and co-workers report the first structure (and a functional study) of a phosphatidylcholine-specific phospholipase C (PC-PLC) from Listeria monocytogenes (Lm), the causative agent of the foodborne disease listeriosis. The authors show compelling evidence to support a model of the synergy between LmPC-PLC activity and the pore-forming activity of pore-forming toxin listeriolysin O (LLO). Several regulation mechanisms are suggested including through oligomerization and inhibition by its pro-peptide. Altogether, this work is providing novel insights into the structure and function of LmPC-PLC and its role in infection by Listeria monocytogenes. The data look compelling and the conclusions justified. I only have a few comments.

Plasticity of the active site (p.11-12): I do not understand how the authors reconcile the unusual position of D55 with its crucial role in catalysis. I recommend that the authors clarify that point. The authors discuss the difference in the conformation of the loops D55-S64 and R75-F83 in MolA and MolB, and also in comparison to Bc and CpPC-PLC. In particular the discussion of the role of these loops in occluding the active site is very interesting. I am surprised though that the authors do not consider, in this discussion, the potential role of these loops in the interaction of LmPC-PLC with membranes. Is it conceivable that these loops would be stabilized in their open state through interactions with membrane lipids? Or is it known that these loops are not involved in membrane binding. If so, which ones are?

We thank the reviewer for these comments. We further clarified this point in the Result paragraph entitled ‘Structural plasticity of the active site and oligomerization regulate the enzymatic activity of LmPC-PLC’, lines 259-268: ‘Despite the spatial dislocation of D55, this residue is still crucial for the enzymatic activity of LmPC-PLC, as the LmPC-PLCD55N mutant was inactive (Fig. 3b) and could play a similar role in catalysis as its counterpart, D55 in BcPC-PLC, where it was suggested to play the role of a general base via a nucleophilic attack on a water molecule bound to Zn1 and Zn. The structural flexibility of the loops S2-T9, D55-S64 and R75-F83 surrounding the active site may thus enable the catalytical effective adjustment of the position of D55 upon saturation of the active site with Zn ions (Zn1) as well as upon binding of the (lipid membrane) substrate. The structural plasticity of the active site cleft in combination with the low affinity for Zn ions and the tendency to oligomerize into enzymatically much weaker oligomers could represent a regulatory mechanism the enzymatic activity of LmPC-PLC.’

The surface of BcPC-PLC is rich in tyrosines and tryptophanes and it has been shown, for BtPI-PLC (Grauffel et al, JACS, 2014) and spider venom GPD-like phospholipases D (SicTox, Moutoussamy et al, PLoS Comp Biol, 2022), that tyrosines and tryptophanes engage in specific interactions with PC lipids (including the formation of aromatic cages around choline groups). The alignment of the Lm and Bc PC-PLC sequences show that several of Bc Tyr and Trp are conserved in the Lm enzyme, including in the loops 55-64 and 75-83. Are these solvent-exposed in the LmPC-PLC?

We thank the reviewer for this comment. Indeed, all three bacterial PC-PLC homologs have surface exposed Tyr and Trp residues, many of which are conserved. Interestingly, most of them are concentrated on one side of the molecule, around the active site cleft, which additionally suggest the orientation of the molecule toward the membrane substrate – with the side containing the active site facing the membrane. This especially true for the region including three surface exposed Tyr residues in the dynamic loops D55-S64 and R75-F83 (Y60, Y61, Y79) that could be involved in membrane binding and thus the stabilization of the dynamic loops. We added another panel on the Supplementary Fig. 8c and marked them additionally on the Supplementary Fig 1, and explained it in the main text, lines 232-238.

In addition, based on the structure of *Bc*PC-PLC with the substrate analogue the choline binding pocket was identified. This consists of three choline binding residues, E4, Y56 and F66, with the latter two forming a π -cation interaction with the choline headgroup (Martin et al., 2000, 10.1016/j.chom.2012.06.010). In *Lm*PC-PLC, only F66 is conserved in the primary structure, however, it occupies a completely different position in *Lm*PC-PLC than in *Bc*PC-PLC. E4 is substituted with D4 in *Lm*PC-PLC and Y56 with H56. However, D4 in *Lm*PC-PLC is located in the flexible loop S2-T9, which is structurally not defined. Moreover, the position of H56 in *Lm*PC-PLC is offset in comparison to Y56 in *Bc*PC-PLC. Whereas the choline binding pocket appears to be formed in *Bc*PC-PLC already in the absence of the substrate, this does not seem to be the case for *Lm*PC-PLC. Previous studies have shown that D4/H56 in *Lm*PC-PLC could be replaced by *Bc*PC-PLC sequence with no harm to enzymatic activity (Zückert et al., 1998, 10.1128/iai.66.10.4823-4831.1998, Huang et al., 2016, 10.1016/j.bbapap.2016.03.008). Therefore, it is unclear if such binding pocket exists in *Lm*PC-PLC. Another possibility is that the pocket is formed upon membrane or substrate binding since *Lm*PC-PLC exhibits high flexibility in the regions that include those residues. We included these considerations in Discussion, lines 516-533, and Supplementary Fig 8d.

The pLDDT scores (per residue) should be reported for the AlphaFold model of the proenzyme, so that the reader can evaluate the quality of the model in the propeptide region. Values below 70 should be treated with caution.

Thank you for this comment, we have added the pLDDT values to (now) Supplementary Fig. 13b and further comment on this in the manuscript (lines 329-330). pLDDT scores do have rather low value in the propeptide region (40 - 60), which was not surprising, as there are no (to our knowledge) models available in PDB that would help build such a model. We agree that the model has poor confidence in the propeptide region and we do not claim at all that this is how the pro-enzyme structure looks). We state clearly in the text that we deal only with the model and not the experimentally determined structure. In the revised version, we also moved the panel with the AF2 model of pro-*Lm*PC-PLC from Fig. 4d to Supplementary Fig. 13a.

We used the AF2 model solely to help us design the peptides to test the inhibitory potential of the propeptide region when added *in trans* to the mature enzyme. Another interesting suggestion from the AF2 model was that of one of the two cysteines from the propeptide and the C143 on the surface of the mature enzyme could form a disulfide bond. Therefore, based on this, we designed the shorter peptide (containing cysteines) in addition to the full propeptide as potential inhibitors of the mature *Lm*PC-PLC. And indeed, we showed that the peptides specifically inhibited *Lm*PC-PLC and that the disulfide bond formation appears to play a role here, at least to some extent (Fig. 4g, h).

**Response to Reviewer #3:**

Given my expertise, I have been primarily invited to comment on the crystallographic content of this manuscript. However, since I also have extensive expertise in bioinorganic and protein chemistry, I have made additional comments related to those aspects of the manuscript.

Crystallography: I only have access to the validation report (not coordinates or maps). The data listed in Table 1 (SI) indicate a resolution cut-off for the data of 2.0 angstroms, however the $I/\sigma I$ in the highest bin = 4.19 (2.072 - 2.0 angstroms), which indicates the data have been artificially truncated and may have been processed to higher resolution. Can the authors revisit the data processing to see if this might be possible?

We thank the reviewer for this comment. We have used the program XDSApp (Krug et al., 2012 10.1107/S0021889812011715), where initially, the optimal resolution cut off at 2.0 Å was determined automatically by the program. We did, however, re-process the data with cutoff at lower resolution shells, but the statistics dropped significantly right after 2.0 Å, so we decided to stick to the suggestion given by the processing algorithm.

The data were collected using X-radiation of wavelength 1.26 angstroms, which is at the Zn K-edge. This is sensible for structure solution, but these data will probably suffer from some radiation damage due to absorption by the Zn atoms in the crystal. A more appropriate method would be to collect a 'native' dataset away from the edge (eg. at 13.0 keV) and use those data for refinement. I would also like to see anomalous difference Fourier electron density maps calculated from data collected at an energy 'below' the Zn-edge. These maps should show a lack of anomalous signal and then confirm the metal atoms unequivocally as Zn. At the moment, the metal atoms in the structure could be Zn, Co, Ni, Cu or Fe, based on the fact that anomalous signal was detected from the data collected at 1.26 angstroms.

We thank the reviewer for these valuable and constructive comments. We agree with the protocol suggested by the reviewer. We did collect data also at the X-ray wavelength of 1.00 Å, however, the dataset on this crystal (or on any other crystal) at 1.26 Å wavelength was better than at 1 Å, so we then decided to also refine using this dataset collected at the wavelength of 1.26 Å. To check for any potential radiation damage, we processed the first half of the set (frames 1-400) and second half (frames 401-800). A summary of data is shown below and it is evident that no significant drop in quality of data was observed through the collection, indicating no significant radiation damage.

(1) Processing of all images (1-800, Dphi = 1°), output (correct.lp)

SUBSET OF INTENSITY DATA WITH SIGNAL/NOISE ≥ -3.0 AS FUNCTION OF RESOLUTION													
RESOLUTION LIMIT	NUMBER OF REFLECTIONS			COMPLETENESS OF DATA	R-FACTOR observed	R-FACTOR COMPARED expected	I/SIGMA	R-meas	CC(1/2)	Anomal Corr	SigAno	Nano	
5.94	30297	2038	2041	99.9%	2.5%	2.6%	30297	91.38	2.6%	100.0*	74*	2.479	816
4.22	55201	3638	3638	100.0%	3.3%	3.2%	55201	75.96	3.4%	100.0*	61*	1.962	1688
3.45	70418	4666	4669	99.9%	4.3%	4.0%	70418	63.32	4.4%	100.0*	46*	1.579	2126
2.99	85715	5596	5596	100.0%	6.6%	6.3%	85715	42.01	6.9%	99.9*	35*	1.354	2586
2.68	92825	6263	6263	100.0%	11.0%	10.9%	92825	25.46	11.4%	99.8*	26*	1.128	2924
2.45	102126	6976	6976	100.0%	17.3%	17.6%	102126	16.58	18.0%	99.5*	20*	0.945	3276
2.27	110052	7570	7579	99.9%	27.5%	28.7%	110052	10.62	28.5%	98.9*	8	0.839	3568
2.12	109399	7912	8086	97.8%	41.0%	41.0%	109399	7.19	42.5%	97.5*	13*	0.854	3710
2.00	116705	8615	8659	99.5%	63.2%	67.5%	116682	4.26	65.7%	93.4*	8	0.800	4081
total	772738	53274	53507	99.6%	8.5%	8.5%	772715	27.07	8.9%	100.0*	23*	1.128	24695

NUMBER OF REFLECTIONS IN SELECTED SUBSET OF IMAGES 785962
NUMBER OF REJECTED MISFITS 10103
NUMBER OF SYSTEMATIC ABSENT REFLECTIONS 468
NUMBER OF ACCEPTED OBSERVATIONS 775391
NUMBER OF UNIQUE ACCEPTED REFLECTIONS 53461

(2) Processing of the first half of images (1-400, Dphi = 1°), output from correct.lp

SUBSET OF INTENSITY DATA WITH SIGNAL/NOISE ≥ -3.0 AS FUNCTION OF RESOLUTION													
RESOLUTION LIMIT	NUMBER OF REFLECTIONS			COMPLETENESS OF DATA	R-FACTOR observed	R-FACTOR COMPARED expected	I/SIGMA	R-meas	CC(1/2)	Anomal Corr	SigAno	Nano	
5.74	16416	2249	2258	99.6%	2.2%	2.3%	16415	69.25	2.3%	100.0*	65*	1.906	906
4.08	30713	4036	4036	100.0%	2.9%	2.8%	30713	59.55	3.1%	100.0*	49*	1.539	1794
3.34	39297	5207	5207	100.0%	4.1%	3.7%	39297	46.71	4.4%	99.9*	34*	1.315	2382
2.89	46948	6139	6141	100.0%	6.7%	6.2%	46948	29.28	7.2%	99.8*	23*	1.137	2843
2.59	50654	6994	6994	100.0%	12.0%	11.6%	50652	16.51	12.9%	99.5*	18*	0.976	3272
2.36	56091	7703	7703	100.0%	17.7%	17.8%	56089	11.30	19.0%	98.9*	6	0.866	3625
2.19	58288	8287	8367	99.0%	29.8%	30.3%	58287	6.78	32.3%	96.8*	6	0.803	3894
2.05	61932	8992	9000	99.0%	42.8%	45.4%	61918	4.50	46.3%	93.7*	4	0.756	4259
1.93	61607	9587	9594	99.9%	67.8%	75.1%	61597	2.56	73.8%	83.5*	0	0.703	4561
total	421946	59194	59300	99.8%	8.0%	8.0%	421916	19.31	8.6%	99.9*	14*	0.971	27536

NUMBER OF REFLECTIONS IN SELECTED SUBSET OF IMAGES 426678
NUMBER OF REJECTED MISFITS 3944
NUMBER OF SYSTEMATIC ABSENT REFLECTIONS 228
NUMBER OF ACCEPTED OBSERVATIONS 422506

(3) Processing of the second half of images (401-800, Dphi = 1°), output from correct.lp

SUBSET OF INTENSITY DATA WITH SIGNAL/NOISE ≥ -3.0 AS FUNCTION OF RESOLUTION													
RESOLUTION LIMIT	NUMBER OF REFLECTIONS			COMPLETENESS OF DATA	R-FACTOR observed	R-FACTOR COMPARED expected	I/SIGMA	R-meas	CC(1/2)	Anomal Corr	SigAno	Nano	
5.86	15494	2113	2119	99.7%	2.2%	2.3%	15493	68.12	2.4%	100.0*	68*	1.925	849
4.16	28786	3799	3799	100.0%	3.0%	2.9%	28786	57.68	3.2%	99.9*	51*	1.563	1683
3.40	36846	4914	4918	99.9%	4.1%	3.7%	36846	46.19	4.4%	99.9*	35*	1.324	2239
2.95	44427	5806	5806	100.0%	6.7%	6.2%	44427	29.28	7.2%	99.8*	24*	1.153	2689
2.64	47712	6556	6558	100.0%	11.9%	11.5%	47711	16.60	12.8%	99.5*	23*	0.996	3059
2.41	53333	7316	7316	100.0%	18.2%	18.4%	53332	10.96	19.6%	98.8*	5	0.849	3440
2.23	55521	7810	7882	99.1%	31.0%	31.4%	55520	6.62	33.5%	96.8*	8	0.815	3658
2.09	58811	8455	8458	100.0%	42.4%	45.9%	58801	4.50	45.8%	94.0*	-1	0.726	4006
1.97	61036	9045	9056	99.9%	72.0%	78.3%	61031	2.59	78.1%	83.9*	4	0.714	4300
total	401966	55814	55912	99.8%	8.1%	8.1%	401947	19.03	8.8%	99.9*	15*	0.974	25923

NUMBER OF REFLECTIONS IN SELECTED SUBSET OF IMAGES 406424
NUMBER OF REJECTED MISFITS 3685
NUMBER OF SYSTEMATIC ABSENT REFLECTIONS 240
NUMBER OF ACCEPTED OBSERVATIONS 402499
NUMBER OF UNIQUE ACCEPTED REFLECTIONS 55886

In addition, we agree that we should have provided more evidence for the identity of atomic species in the active site. To address this, we first determined presence of the metal ions in the purified *LmPC-PLC* using ICP-MS and ICP-OES analyses (please see the revised text, the new paragraph in results: ‘The active site of recombinant *LmPC-PLC* contains Zn and Fe ions’, starting at the line 156). The only species enriched in the protein samples were Zn and Fe, Fe being much more abundant than Zn (Supplementary Fig. 6c). Based on these measurements we determined the ratio of Fe ion to one protein molecule as 1.43 and 0.01 for Zn. This information is now displayed on Supplementary Fig. 5. Low abundance of Zn ions in the active site is in line with our biochemical data showing that Zn is required for activity as we showed in Fig. 2a (and also, Huang et al., 2016, 10.1016/j.bbapap.2016.03.008).

We then collected and processed data from a crystal from the same batch as deposited structure at wavelengths 1.33 Å (below Zn edge) and 1.26 Å (close to the higher energy side of the Zn-edge). For the position of the metal ion 1, we observed a signal in the anomalous difference map of the data collected at 1.26 Å X-ray wavelength, which was not present in the anomalous difference map from 1.33 Å data (Supplementary Fig. 6a, b). This suggested that the metal ion at the position 1 was Zn. On the other hand, anomalous difference map from the data collected at 1.33 Å showed strong peaks at metal ion position 2 and especially at position 3, indicating the presence of metal ions other than Zn (Supplementary Fig. 6a, b). Based on ICP-OES and ICP-MS analyzes we concluded, that the metal ions at position 2 and 3 were Fe ions. Interestingly, while the strength of the anomalous signal at the metal ion position 3 (Fe3) was comparable between the anomalous difference maps of 1.26 Å and 1.33 Å data, the anomalous signal at metal ion position 2 was significantly weaker in the anomalous difference map of the data collected at 1.33 Å (Supplementary Fig. 6a, b), suggesting that the metal ion position 2 may contain Fe ions in some molecules and Zn ions in others. It should be noted that no Zn or Fe ions were added during *LmPC-PLC* purification or crystallization. With the submission of the revised manuscript, we have also included the requested files (file name Reviewer3_files_Petrisic.zip. It includes deposited PDB file for the crystal structure of *LmPC-PLC* and the corresponding MTZ file. It also includes two MTZ files for the anomalous differences maps, of the data collected at 1.26 Å and 1.33 Å X-ray wavelength).

Ratios of bound ion metals to protein as shown in the Supplementary Fig. 5c agree with the occupancies of Fe ions in the crystal structure (Supplementary Table 2). Low ratio of Zn bound to the protein explains a weak anomalous signal as well as low occupancy of Zn in the actives site (0.44 in molA to 0 in mol B). We added all this additional information into the text of the manuscript and Supplementary Fig. 6. Metal ions were also shown accordingly in all figures. Relatively low affinity for the Zn ion was shown for *LmPC-PLC* in previous studies, with K_d determined around 60 μM (Huang et al., 2016, 10.1016/j.bbapap.2016.03.008).

I find the description of the Zn coordination lacking in this manuscript, both in text and in Figure 3a (inset). The ‘reorientation’ of residue D55 seems to correlate with decreased occupancy of the Zn1 site, however the other putative ligands seem to be in place. There is no discussion on what interactions might be contributing to the altered orientation of D55, therefore, it is very difficult to understand the differences between this structure and homologues. D55 is obviously crucial for activity, which makes this more perplexing. The activity of the enzyme is measured in the presence of various divalent cations (Fig 2a), but there doesn’t seem to be an attempt to determine the metal:protein stoichiometry (eg. by ICP-MS) and/or affinity (ITC). Is the Zn1 site occupancy lower than Zn2 and Zn3 in the protein in solution or only in the crystal?

We thank the reviewer for this comment, and part of it was already addressed in the previous paragraph. The occupancy of the Zn site is low both in solution and in the crystal, as explained in the paragraph above.

With regards to D55, as already explained in the answers to the reviewer 2, the different position of D55 could indeed be due to the low occupancy of Zn1 site. Moreover, the loops building half of the active site are flexible, including the containing D55 (residues D55-S64), as it can be observed only in molB. Please see our explanation to similar question of the reviewer 2, pages 7-8 of this document. In addition, we have included Supplementary Fig. 7 that shows coordination of ions in the active site as suggested. Fe ions have octahedral coordination while the coordination of Zn ion is tetrahedral.

The data presented in Figure 1 and S2 regarding the estimates molecular weight of the LmPC-PLC protein are confusing at best. Apparently, the dotted line in Fig 1a represents an elution volume of 66 mL, which corresponds to a 28 kDa globular protein, yet peaks 1 (close to the 28 kDa mark) and 2 (eluting after the 28 kDa mark) are described as oligomers. Peak 3 elutes well after this mark and therefore appears to have a molecular weight of around 10 kDa, however this is not estimated. This section needs to be revisited and revised. A preparative column (Fig 1) should never be used for the estimation of molecular weight - at least an analytical column should be employed and more preferably a technique such as MALLS or AUC. At the moment, the analysis is not well described. It is also used to support the crystal structure as consisting of 'monomers' even though the crystal packing shows associations between the protein units in the crystal.

We thank the reviewer for this comment, we agree that we should have described this part more clearly. The dashed line in SEC graph in Fig. 1a marks the position, where we expected that LmPC-PLC monomer should elute (28 kDa mass according to the calibration standards, see Supplementary Fig. 3b, where a dashed line was added to illustrate this more clearly. The red rectangle shows that LmPC-PLC is an outlier). In the first section of Results, we have explained that LmPC-PLC eluted from the column much later than expected. We also noticed several absorbance peaks at the elution volume where (according to standards) a 10 kDa protein should elute, but those peaks contained a 28 kDa protein (as determined by SDS-PAGE). This shift from 28 kDa to an apparent 10 kDa could be due to the nonspecific interaction of the protein with the column matrix. This indicates either a different hydrodynamic volume of LmPC-PLC compared with the protein standard of similar molecular mass or interactions between LmPC-PLC and the column matrix despite high salt concentration in the mobile phase. We added this explanation to the first Result section of the revised manuscript (starting with the line 99).

However, we do agree with the reviewer that the oligomerization and the delay in elution phenomenon should be examined by SEC-MALS, using analytical column. We have performed SEC-MALS on using (Tricorn 10/300 Superdex 75 GL) and the results can be found on Supplementary Fig. 2d. We detected LmPC-PLC mainly in monomeric and dimer forms and partly also in the form of higher order oligomers. The corrections are now included in the paragraph entitled 'LmPC-PLC tends to oligomerize, but only the monomer is enzymatically active', that starts with the line 99.

With regards to the dimer of LmPC-PLC molecules present in the asymmetric unit. We show their relative orientation on Fig. 2b and say in the lines 205-208 that their interacting area is only 4.4 %. This evidence suggests that the active unit is monomeric. We also show that oligomeric species are basically devoid of enzymatic activity (Fig. 1c).

REVIEWERS' COMMENTS

Reviewer #1 (Remarks to the Author):

The authors have substantially addressed all my comments/questions and I am satisfied with most of them. Only a few points require clarification before publication of the study.

L25. The “W1” is not clear for general audience at the first sight in the abstract. Would it be possible to make it more clear here?

L132-134. Based on the results shown in Fig 1e, both the double mutants, including C143S+C168K, showed reduced formation of oligomers. Do the authors intend to have C143S+C168K in the description or exclude it for specific reasons?

L262-268. Since the authors did not design further experiments to test the potential regulatory mechanism of the flexible loops surrounding the active site, would it be better to have this part in discussion?

L353-357. While the score for the prediction is low, the evidence for the disulfide bond formation is rather weak and merely indicates the involvement of C143. I would suggest to tone down in the result and have this hypothesis in discussion.

L535-536. I guess the authors want to highlight the potential interaction from the AlphaFold2 prediction. It is better to be more clear here if possible.

In the supplementary material

Supp. fig 1. L32-33.

I think the reason why the authors provide S at the two C sites of LmPC-PLC is because they cite the PDB-ID, where the C143S+C168S mutant was used for structure analysis. However, in the figure legends the authors stated that two orange stars show the positions of the two unique cysteine residues, which are not seen in the figure. It is better to provide “C” in the alignments, or stated more clear in the figure legends for clarification.

Supp. fig 11. L131-132.

I guess it is “mM” for the concentration of NaCl. Please check and provide it.

Supp. fig 17.

SM in the membrane appeared to specifically reduce the damage by BcPC-PLC as compared to the other PLCs. It would be great to provide an explanation for this as well as the importance of having the result of SM—containing membrane.

Reviewer #2 (Remarks to the Author):

I would like to thank the authors for taking all my comments and questions into account, and providing detailed answers.

All my comments and questions have been addressed satisfactorily and I strongly recommend this manuscript for publication.

Reviewer #3 (Remarks to the Author):

I am entirely satisfied that the revised manuscript addresses all the points made in my review.

Ljubljana, September 6th, 2023**Point-by-point response to the reviewers' for the final revision for Nature Communications manuscript NCOMMS-23-11000A, by Petrišič et al.**

For clarity, the reviewers' comments are in blue and in *italics*. Authors' responses are in regular black text.

Response to Reviewer #1:

The authors have substantially addressed all my comments/questions and I am satisfied with most of them. Only a few points require clarification before publication of the study.

L25. The "W1" is not clear for general audience at the first sight in the abstract. Would it be possible to make it more clear here?

We thank the reviewer for the suggestion, L26 now reads (in bold below):
'...including the invariant position of the N-terminal tryptophan (W1),...'

L132-134. Based on the results shown in Fig 1e, both the double mutants, including C143S+C168K, showed reduced formation of oligomers. Do the authors intend to have C143S+C168K in the description or exclude it for specific reasons?

We thank the reviewer for this comment. We agree that we should have included this double mutant as it also showed reduced formation of oligomers.

The text in (now) lines L133-L134:

'*LmPC-PLC*^{C168S}, *LmPC-PLC*^{C143S+C168S} and *LmPC-PLC*^{C143S+C168K} showed slightly reduced formation of oligomers...'

We stated (already before) in the last sentence of this paragraph, now L139, 'Of all the constructs, *LmPC-PLC*^{C143S+C168S} had the highest solubility and successfully produced crystals.'

So, we chose the C143S+C168S mutant for crystallographic studies because it appeared to have the least oligomer formation on SDS-PAGE (Fig. 1e), formed fewer aggregates upon purification, and behaved notably better during concentration than other forms. Therefore, with regards to structure determination of PC-PLC, the C143S+C168K mutant was not mentioned anymore, but was included in several other experiments, as shown on figures.

L262-268. Since the authors did not design further experiments to test the potential regulatory mechanism of the flexible loops surrounding the active site, would it be better to have this part in discussion?

We thank the reviewer and agree with this suggestion. Thus, we have removed these sentences from the result section, stating:

(1)

‘The structural flexibility of the loops S2-T9, D55-S64 and R75-F83 surrounding the active site may thus enable the catalytical effective adjustment of the position of D55 upon saturation of the active site with Zn ions (Zn1) as well as upon binding of the (lipid membrane) substrate.’

And

(2) ‘The structural plasticity of the active site cleft in combination with the low affinity for Zn ions and the tendency to oligomerize into enzymatically much weaker oligomers could represent a regulatory mechanism the enzymatic activity of *LmPC-PLC*.’

We agree that this part fit better in the Discussion section. Since a statement similar to (1) has already been made in discussion before (before L517-521 in PDF), we removed the statement (1) from the text altogether.

The statement in discussion with as similar message is (now L429-433):

‘Therefore, the shifted position of D55 in *LmPC-PLC* could be due to the low Zn1 occupancy and flexibility of the active site loops D55-S64, S2-T9, K57-Y61 and N77-L80. *LmPC-PLC* may thus switch between an enzymatically inactive and an active state, triggered by binding of a lipid membrane substrate and saturation of the active site with Zn ions.’

The statement (2) was moved from results to discussion (now L445-448).

L353-357. While the score for the prediction is low, the evidence for the disulfide bond formation is rather weak and merely indicates the involvement of C143. I would suggest to tone down in the result and have this hypothesis in discussion.

We thank the reviewer for these comments. We agree that we do not have firm evidence for the disulfide bond between C143 and the propeptide cysteine. We have toned down this in the result section as well as moved the hypothesis to discussion (for the latter, please, see the answer to the next comment).

Now: L291-296

‘This suggests that C143, located near the active site, may be involved in (pro)peptide-induced inhibition, as also indicated by the AF2 model of pro-*LmPC-PLC* (Supplementary Fig. 13a). However, the C143S mutation does not completely prevent inhibition of *LmPC-PLC*^{C143S} or *LmPC-PLC*^{C143S+C168S} by either peptide, suggesting the role of other residues of the active site cleft and the propeptide in successful inhibition of the enzymatic activity of *LmPC-PLC*.’

L535-536. I guess the authors want to highlight the potential interaction from the AlphaFold2 prediction. It is better to be more clear here if possible.

Here we wanted to highlight the specificity of the propeptide for *LmPC-PLC*, based on the inhibition studies using the propeptide *in trans*, rather than the AlphaFold2 model.

For clarity, we have rewritten the mentioned paragraph in the Discussion (now L449-453):

‘Fourth, we have shown that *LmPC-PLC* can be inhibited by addition of its propeptide *in trans*, and the inhibition is largely specific to *LmPC-PLC*, as no significant inhibition was observed with *Bc-* or *CpPC-PLC*. This is enabled by specific a sequence of both the propeptide and the mature part of the enzyme, which may include the disulfide bond formation between C143 in the mature part of *LmPC-PLC* and one of the cysteines from the propeptide’.

In the supplementary material

Supp. fig 1. L32-33.

*I think the reason why the authors provide S at the two C sites of *LmPC-PLC* is because they cite the PDB-ID, where the C143S+C168S mutant was used for structure analysis. However, in the figure legends the authors stated that two orange stars show the positions of the two unique cysteine residues, which are not seen in the figure. It is better to provide “C” in the alignments, or stated more clear in the figure legends for clarification.*

We thank the reviewer for this comment. We have changed S143 and S168 to C as suggested.

Supp. fig 11. L131-132.

I guess it is “mM” for the concentration of NaCl. Please check and provide it.

Thank you for noticing the typo. We have added ‘mM’ as it was deleted by mistake.

Supp. fig 17.

*SM in the membrane appeared to specifically reduce the damage by *BcPC-PLC* as compared to the other PLCs. It would be great to provide an explanation for this as well as the importance of having the result of SM—containing membrane.*

The introduction of SM into our model membrane systems was done to demonstrate that the effect of PC-PLCs on LLO binding and pore formation is also observed in the presence of SM. Because SM is ubiquitous in the animal plasma membrane, it was important to include it in our studies.

We added the following sentence describing our motivation in the main text (now L330-332):

‘SM is ubiquitous in the animal plasma membrane, which typically contains about 15 mol % of this lipid⁵⁴. Therefore, we have included SM in our lipid membrane models.’

SM in membranes indeed decreased the activity of *BcPC-PLC* (Supplementary Fig. 17b), but *BcPC-PLC* still largely stimulates LLO binding (Fig. 5e).

We added this in (now) L336-338 ‘Interestingly, although the presence of SM in lipid membranes reduces the enzymatic activity of *BcPC-PLC*, the residual activity of *BcPC-PLC* still strongly stimulates the binding of LLO to the SM-containing membrane.’

Reviewer #2:

I would like to thank the authors for taking all my comments and questions into account, and providing detailed answers.

All my comments and questions have been addressed satisfactorily and I strongly recommend this manuscript for publication.

Reviewer #3

I am entirely satisfied that the revised manuscript addresses all the points made in my review.